# Reactive Oxygen Species and Folate Receptor-Targeted Nanophotosensitizers Composed of Folic Acid-Conjugated and Poly(ethylene glycol)-Chlorin e6 Tetramer Having Diselenide Linkages for Targeted Photodynamic Treatment of Cancer Cells

**DOI:** 10.3390/ijms23063117

**Published:** 2022-03-14

**Authors:** Seong-Won Yang, Young-IL Jeong, Min-Suk Kook, Byung-Hoon Kim

**Affiliations:** 1Department of Ophthalmology, College of Medicine, Chosun University, Gwangju 61452, Korea; smarteyes@hanmail.net; 2Department of Dental Materials, School of Dentistry, Chosun University, Gwangju 61452, Korea; jeongyi@chosun.ac.kr; 3Research Institute of Convergence of Biomedical Sciences, Pusan National University Yangsan Hospital, Gyeongnam 50612, Korea; 4Department of Maxillofacial Surgery, School of Dentistry, Chonnam National University, Gwangju 61186, Korea

**Keywords:** photodynamic therapy, reactive oxygen species, retinoblastoma, nanophotosensitizers, drug targeting

## Abstract

Folic acid-conjugated nanophotosensitizers composed of folic acid (FA), poly(ethylene glycol) (PEG) and chlorin e6 (Ce6) tetramer were synthesized using diselenide linkages for reactive oxygen species (ROS)- and folate receptor-specific delivery of photosensitizers. Ce6 was conjugated with 3-[3-(2-carboxyethoxy)-2,2-bis(2-carboxyethoxymethyl)propoxy]propanoic acid (tetra acid, or TA) to make Ce6 tetramer via selenocystamine linkages (TA-sese-Ce6 conjugates). In the carboxylic acid end group of the TA-sese-Ce6 conjugates, FA-PEG was attached again using selenocystamine linkages to make FA-PEG/TA-sese-Ce6 conjugates (abbreviated as FAPEGtaCe6 conjugates). Nanophotosensitizers were fabricated by a dialysis procedure. In the morphological observations, they showed spherical shapes with small diameters of less than 200 nm. Stability of the aqueous FAPEGtaCe6 nanophotosensitizer solution was maintained (i.e., their particle sizes were not significantly changed until 7 days later). When H_2_O_2_ was added to the nanophotosensitizer solution, the particle size distribution was changed from a monomodal pattern to a multimodal pattern. In addition, the fluorescence intensity and Ce6 release rate from the nanophotosensitizers were also increased by the addition of H_2_O_2_. These results indicated that the nanophotosensitizers had ROS-sensitive properties. In an in vitro cell culture study, an FAPEGtaCe6 nanophotosensitizer treatment against cancer cells increased the Ce6 uptake ratio, ROS generation and light-irradiated cytotoxicity (phototoxicity) compared with Ce6 alone against various cancer cells. When the folic acid was pretreated to block the folate receptors of the Y79 cells and KB cells (folate receptor-overexpressing cells), the intracellular Ce6 uptake, ROS generation and thereby phototoxicity were decreased, while the MCF-7 cells did not significantly respond to blocking of the folate receptors. These results indicated that they could be delivered by a folate receptor-mediated pathway. Furthermore, an in vivo pulmonary metastasis model using Y79 cells showed folate receptor-specific delivery of FAPEGtaCe6 nanophotosensitizers. When folic acid was pre-administered, the fluorescence intensity of the lungs was significantly decreased, indicating that the FAPEGtaCe6 nanophotosensitizers had folate receptor specificity in vitro and in vivo. We suggest that FAPEGtaCe6 nanophotosensitizers are promising candidates for a targeted photodynamic therapy (PDT) approach against cancer cells.

## 1. Introduction

Photodynamic therapy (PDT) has been extensively investigated in the last two decades because it is regarded as a safe candidate for the treatment of cancer patients [1,2,3,4,5]. PDT and its output are normally composed of a photosensitizer, light and oxygen species, and furthermore, photosensitizers are only activated in a specific wavelength of light and then produce reactive oxygen species (ROS) (i.e., PDT produces excessive ROS in the site of irradiation and then inhibits the viability of disease cells) [6]. Therefore, PDT has a toxic reaction only in the site of light irradiation, which means it is able to minimize any adverse effects against normal cells or tissues [3,4,7,8]. For example, Leroy et al. reported that 5-aminolevulinic acid (5-ALA)-based PDT for brain cancer is efficient and safe for high-grade glioma [8]. They argued that glioma patients receiving 5-ALA PDT recorded longer survival times than those without PDT. Furthermore, PDT induces a response rate higher than 90% and no local recurrence in oral squamous carcinoma cells [8]. In spite of these advantages, some drawbacks to photosensitizers still limit the clinical application of PDT. PDT application is hard to apply to cancers located in deep tissues because photosensitizers have no cytotoxic reactions in the absence of light irradiation (i.e., the light irradiation depth is below 15 mm), and PDT alone has efficacies against cancers in the epithelial region or mucous layer [9]. Furthermore, conventional photosensitizers have no specificity against tumors. Thus, they are frequently freely dispersed in the body, and this problem induces long periods of sunshade [3]. To solve these problems, various kinds of photosensitizers have been investigated [2,5,10,11,12,13]. For example, Mono-*L*-aspartyl chlorin e6 (NPe6) represented a favorable photosensitivity (i.e., the photosensitivity of NPe6 was resolved within 1 week) [5]. In particular, nanophotosensitizers, which are photosensitizer-incorporated nano-sized carriers, have been suggested as an ideal candidate for PDT [12,13,14]. Various biocompatible polymers or nanomaterials such as natural polysaccharides, dendrimers, iron oxide nanoparticles and synthetic polymers have been investigated to fabricate nanophotosensitizers [12,13,14,15,16]. Chin et al. reported that poly(vinylpyrrolidone) (PVP)-Ce6 formulations accelerate the tumor accumulation of photosensitizers and show fast clearance from the skin of nude mice [11]. Poly-cationic polysaccharides such as chitosan can be used to fabricate Ce6-encapsulated nanophotosensitizers [12]. They argued that nanophotosensitizers can be formed by an ion complex formation between water-soluble chitosan and Ce6 (ChitoCe6). The ChitoCe6 nanophotosensitizers showed superior accumulation in the tumor tissue of nude mice compared with free Ce6 and improved PDT efficacy against gastrointestinal cancer cells [12].

On the other hand, nanocarrier-based drug delivery systems have been spotlighted in last several decades due to their superiority in drug targeting issues [17,18,19,20,21]. Due to their small size, nanocarriers are believed to be an ideal candidate for tumor targeting [17,18]. Since the surface of nanocarriers such as polymeric nanoparticles can be easily decorated with targeting moieties such as monoclonal antibodies, they can then be specifically delivered to the tumor with a minimization of side effects against normal cells [17,18]. Furthermore, nanocarriers based on polymeric nanoparticles or micelles can be designed to stimulate of tumor tissues and then deliver the anticancer drug to the tumor tissues specifically [18,19,20,21]. For example, Lei et al. reported that polymeric micelles having pH and temperature dual-responsive properties accelerate the release rate of anticancer drugs at an acidic pH and a higher temperature [20]. From these points of view, stimuli-responsive nanocarriers have great potential in tumor targeting, because the physiological properties of the tumor microenvironment are quite different from their normal counterparts (i.e., tumor tissues are characterized as having an acidic pH, abnormal redox status, abundant extra- or intra-cellular enzymes and elevated expression of various molecular receptors) [22,23,24,25]. Therefore, the abnormal status of a tumor microenvironment provides a tumor targeting opportunity for polymeric nanocarrier-based drug delivery [26]. Interestingly, these statuses for tumor microenvironments can be applied in the design of a nanoparticle architecture sensitive to the ROS level, since the intracellular ROS levels in tumor tissues are significantly higher than those of their normal counterpart [27,28]. For example, Lee et al. reported that chitosan cross-linked with a diselenide group and conjugated with acetyl histidine shows dual sensitivity against the pH and ROS [28]. They argued that nanoparticles sensitively liberate anticancer drugs through an acidic pH and ROS. Those features of nanoparticles accelerate the tumor accumulation of nanoparticles and then efficiently inhibit the metastasis of cancer cells. Furthermore, molecular receptors such as folate or CD44 receptors enable nanoparticles to target cancer cells in a receptor-specific manner [29,30]. Lee et al. reported that folic acid-decorated nanoparticles deliver anticancer agents in a folate-specific manner in in vitro cell culture and in vivo tumor xenograft models [29]. In our previous report, hyaluronic acid (HA) as a hydrophilic segment and hyperbranched Ce6 as a lipophilic segment formed nano-dimensional carriers as nanophotosensitizers through a self-assembling process (i.e., hyperbranched Ce6 formed the inner core of the nanophotosensitizers and the HA exposed on the surface of the nanophotosensitizers) [30]. They argued that nanophotosensitizers composed of HA-hyperbranched Ce6 conjugates via disulfide linkages represent the CD44-specific and redox-sensitive delivery capacity of Ce6’s capacity against tumors [30]. Those nanophotosensitizers showed CD44-specific imaging and PDT efficacy against U87MG glioblastoma cells. Sun et al. also reported that folic acid-decorated nanocarriers show folate-receptor-mediated targeting and permit MRI-guided imaging of tumors [14].

In this study, we synthesized folic acid-conjugated poly(ethylene glycol)/Ce6 tetramer (abbreviated as FAPEGtaCe6) conjugates using 3-[3-(2-carboxyethoxy)-2,2-bis(2-carboxyethoxymethyl)propoxy]propanoic acid (tetra acid, or TA) and fabricated nanophotosensitizers. Tetra acid (TA) was used to synthesize the Ce6 tetramer via diselenide linkages, since diselenide linkages can be disintegrated by ROS [27,28]. Their targeting efficiency against a folate receptor and ROS-sensitivity against cancer cells were studied using Y79 retinoblastoma cells, KB epithelial carcinoma cells and MCF-7 human breast cancer cells. We characterized the physicochemical and biological properties of FAPEGtaCe6 nanophotosensitizers in vitro and in vivo.

## 2. Results

### 2.1. Characterization of FAPEGtaCe6 Conjugates

To synthesize FAPEGtaCe6 conjugates, Ce6 was attached to the four carboxyl groups of TA via the diselenide group to make a Ce6 tetramer as shown in Figure 1a. Ce6 was primarily conjugated with selenocystamine to endow ROS sensitivity to the conjugates (Appendix Aa). Ce6-selenocystamine conjugates were conjugated again with TA to produce TA-sese-Ce6 conjugates as shown in Figure 1a. Appendix A shows the specific peaks of Ce6 and TA (i.e., the specific peaks of Ce6 were confirmed at 1–10 ppm (Appendix Aa)), and the peaks of the ethylene protons of selenocystamine were confirmed at 2.8–3.4 ppm. Furthermore, the specific peaks of TA (ethylene proton) were confirmed between 2 and 4 ppm, as shown in Appendix Aa.

Otherwise, FA was attached to the amine group of bifunctional PEG as shown in Appendix Ab. After that, the remaining carboxylic acid of the bifunctional PEG was attached with selenocystamine again to produce FA-PEG-sese conjugates (Appendix Ab). As shown in Appendix Ab, the ethylene protons of PEG were confirmed at 3.4–3.7 ppm, while the specific peaks of FA were confirmed at 1 and 9 ppm. The ethylene group of selenocystamine was confirmed at 2.8–3.4 ppm. Next, four equivalents of the FA-PEG-sese conjugates were conjugated to the carboxyl group of the TA-sese-Ce6 conjugates as shown in Figure 1b. Figure 1c also shows that the ^1^H NMR spectra showed all of the components of the FAPEGtaCe6 conjugates, the specific peaks of Ce6, selenocystamine, FA, TA and PEG. These results indicate that the FAPEGtaCe6 conjugates were successfully synthesized. Table 1 shows the characteristics of FAPEGtaCe6.

The theoretical contents were calculated as 16.8% (*w*/*w*), and the experimental content of Ce6 was 16.2% (*w*/*w*), indicating that the experimental content of Ce6 was not significantly different from that of the theoretical value.

### 2.2. Fabrication and Characterization of Nanophotosensitizers

FAPEGtaCe6 nanophotosensitizers were fabricated by a dialysis procedure. The ultraviolet-visible (UV-VIS) absorption spectra of Ce6 alone and the FAPEGtaCe6 nanophotosensitizers were measured as shown in Figure 2a. As this figure shows, the FAPEGtaCe6 nanophotosensitizers represented similar UV absorption spectra between 600 and 700 nm in DMSO. Furthermore, a specific peak at 664 nm was also observed both for Ce6 alone and the FAPEGtaCe6 nanophotosensitizers, indicating the intrinsic absorption properties of Ce6. The morphology and particle size were measured via transmission electron microscopy (TEM) and a zetasizer as shown in Figure 2b,c. As illustrated in this figure, the nanophotosensitizers of the FAPEGtaCe6 conjugates showed spherical shapes having small sizes of less than 200 nm. The average particle size was 120.1 ± 37.31 nm, as shown in Figure 2c. These results indicate that the FAPEGtaCe6 conjugates could form nanoparticles in an aqueous environment with a monomodal distribution pattern. Furthermore, the aqueous stability of the FAPEGtaCe6 nanophotosensitizers was maintained for at least 7 days (i.e., the mean particle sizes were not significantly changed for 7 days, even though their average particle sizes were slightly increased, as shown in Figure 2d). Furthermore, neither large aggregates nor precipitants were observed even after 7 days. These results indicate that FAPEGtaCe6 nanophotosensitizers can be stored as an aqueous solution for biological applications. The singlet oxygen (SO) generation efficiency of the nanophotosensitizers was evaluated in aqueous conditions using an SOSG reagent as shown in Figure 2e. The fluorescence intensity of the SOSG was gradually increased according to the increase in irradiation time of both Ce6 alone and the FAPEGtaCe6 nanophotosensitizers. The fluorescence intensity of the nanophotosensitizers in particular was higher than that of Ce6 alone. Furthermore, the fluorescence intensity of the nanophotosensitizers was not significantly increased in the absence of light irradiation. These results indicate that the FAPEGtaCe6 nanophotosensitizers efficiently produced and successfully generated SO in the aqueous solution. The ROS sensitivity of the FAPEGtaCe6 nanophotosensitizers was assessed in the presence of H_2_O_2_ as shown in Figure 3, Appendix A. Figure 3, Appendix A shows that the nanophotosensitizers at 0 mM H_2_O_2_ demonstrated results of 117.1 ± 45.18 nm with a monomodal distribution. However, the particle size distribution became broad and multimodal in its pattern when H_2_O_2_ was added. When 2.0 mM H_2_O_2_ was added, the size distribution of the nanophotosensitizers started to become bimodal in phase. In particular, the particle size distribution of the nanophotosensitizers became a bimodal distribution pattern completely with the addition of 10 mM H_2_O_2_, as shown in Figure 3. These results indicate that the FAPEGtaCe6 nanophotosensitizers had ROS-sensitive properties, and they could be disintegrated according to the oxidative stress.

Figure 4 shows the effect of ROS on the changes in the fluorescence emission spectra and the drug release properties of nanophotosensitizers. As shown in Figure 4a, the intensity of the fluorescence emission spectra was gradually increased according to the increase in the H_2_O_2_ concentration. The fluorescence images in Figure 4a also increased according to the H_2_O_2_ concentration. Furthermore, the Ce6 release became faster with the addition of H_2_O_2_, while the Ce6 release became significantly slower in the absence of H_2_O_2_. When H_2_O_2_ was added to the release media, the Ce6 release rate became faster according to the H_2_O_2_ concentration, indicating that the FAPEGtaCe6 nanophotosensitizers had ROS sensitivity, which means the liberation of Ce6 could be controlled by the oxidative stress in the biological system.

### 2.3. Cell Culture and PDT Study In Vitro

The PDT efficacy of the FAPEGtaCe6 nanophotosensitizers was evaluated with Y79 retinoblastoma cells, MCF-7 human breast cancer cells and KB human epithelial carcinoma cells as shown in Figure 5, Figure 6, Figure 7, Figure 8 and Figure 9. Furthermore, dark toxicity as an intrinsic cytotoxicity against normal cells was assessed with ARPE-19 human retinal pigment epithelial cells in vitro as shown in Figure 6. Prior to assessing the PDT efficacy, the Ce6 uptake ratio was evaluated in vitro as shown in Figure 5. As shown in Figure 5a, the Ce6 uptake ratio for both the free Ce6 and FAPEGtaCe6 nanophotosensitizers was gradually increased according to the increase in Ce6’s concentration. In particular, the FAPEGtaCe6 nanophotosensitizers represented a two or three times higher uptake ratio of all cell types, such as Y79 cells (Figure 5a), MCF-7 (Figure 5b) and KB cells (Figure 5c). Figure 5d shows that treatment with nanophotosensitizers represented a significantly higher red fluorescence intensity in the KB cells than that of the free Ce6. These results indicate that the nanophotosensitizers had a higher efficacy in the intracellular uptake ratio against cancer cells.

Prior to performing the PDT study, the effect of a light dose against the Y79 cells was evaluated. As shown in Appendix A, the viability of the Y79 cells gradually decreased according to the increase in the light dose. The viability of the Y79 cells was not practically affected until 1.5 J/cm^2^ (i.e., the cell viability was higher than 90% at 1.5 J/cm^2^ and then gradually decreased until 20 J/cm^2^). However, the temperature was also raised when the light dose was higher than 5 J/cm^2^, and this may have affected the viability of the cells. To minimize this, 2 J/cm^2^ was used for the PDT study.

The dark toxicity of the Ce6 and FAPEGtaCe6 nanophotosensitizers was evaluated in vitro using noncancerous normal cell lines (ARPE-19 cells (Figure 6a) and HaKaT cells (Figure 6b)) and cancer cell lines (Y79 cells (Figure 6c), MCF-7 cells (Figure 6d) and KB cells (Figure 6e)). As shown in Figure 6a,b, the viability of the ARPE-19 cells and HaKat cells stayed higher than 80% until a 5.0-µg/mL Ce6 concentration in both the free Ce6 and FAPEGtaCe6 nanophotosensitizers. These results indicate that the FAPEGtaCe6 nanophotosensitizers had no significant cytotoxicity against normal cells until a 5.0-µg/mL Ce6 concentration, just like with the free Ce6. In the absence of light irradiation, the FAPEGtaCe6 nanophotosensitizers also had low cytotoxicity against cancer cells such as Y79 cells (Figure 6c), MCF-7 cells (Figure 6d) and KB cells (Figure 6e). Ce6 treatment especially showed that the viability of the Y79, MCF-7 and KB cells was decreased to less than 80% at a 5-µg/mL Ce6 concentration, while the nanophotosensitizer treatment still remained at a cell viability higher than 80%. These results indicate that the nanophotosensitizers had a lower dark toxicity and no acute cytotoxicity against cancer cells in the absence of light irradiation.

Furthermore, the intracellular uptake of the Ce6 or nanophotosensitizers was gradually increased according to the treatment time as shown in Appendix Aa,b (i.e., when the Ce6 or nanophotosensitizers were exposed to cells for 4 h, the intracellular uptake ratio increased both for Ce6 alone and the nanophotosensitizers, as shown in Appendix Ab). In particular, the intracellular uptake ratio of Ce6 alone for 4 h was increased more than two times compared with that at 1.5 h, while the nanophotosensitizer treatment’s intracellular uptake ratio increased 30% compared with that at 1.5 h. These results indicated that the Ce6 uptake rate of the nanophotosensitizers was higher and faster than that of Ce6 alone, even though the Ce6 uptake ratio of Ce6 alone also gradually increased according to the course of time.

Figure 7 shows the effect of the FAPEGtaCe6 nanophotosensitizers on the ROS generation and PDT efficacy. As shown in Figure 7(Aa–Ac), treatment with Ce6 alone and nanophotosensitizers induced a gradual increase in ROS generation in all of the Y79 cells (Figure 7(Aa)), MCF-7 cells (Figure 7(Ab)) and KB cells (Figure 7(Ac)). In particular, ROS generation of nanophotosensitizers in the Y79 cells, MCF-7 cells and KB cells was 3.2 times, 3.1 times and 3.4 times higher than those of Ce6 alone, respectively. These results indicate that the nanophotosensitizers had a higher efficacy in intracellular delivery and ROS generation in cancer cells. As expected, the nanophotosensitizers revealed increased phototoxicity against all cancer cells, as shown in Figure 7(Ba–Bc). The free Ce6 revealed a relatively lower or negligible phototoxicity until a Ce6 concentration of 1.0 µg/mL (i.e., the viability of the cancer cells remained higher than 80% at a Ce6 concentration lower than 1.0 µg/mL). Otherwise, the nanophotosensitizers revealed an increased phototoxicity compared with free Ce6, as shown in Figure 7B (i.e., a decrease in cancer cell viability started from a 0.5-µg/mL Ce6 concentration, and the viability was less than 40% at a 1.0-µg/mL Ce6 concentration). These results indicate that the nanophotosensitizers had improved PDT efficacy against cancer cells compared with the free Ce6.

To assess the folate receptor and ROS-mediated targetability of the nanophotosensitizers, free folic acid (FA) was pretreated to the cells 30 min before nanophotosensitizer treatment as shown in Figure 8 and Figure 9. As shown in Figure 8a, FA pretreatment did not significantly affect to the changes in intracellular Ce6 uptake in the treatment of free Ce6 (i.e., the red fluorescence intensity of the cells for the free Ce6 treatment did not significantly change under pretreatment of FA). However, FA pretreatment significantly decreased the red fluorescence in the nanophotosensitizer treatment. Furthermore, the intracellular uptake ratio of each cell also showed that the free Ce6 treatment was not affected by FA pretreatment, while the nanophotosensitizers significantly changed in fluorescence intensity, as shown in Figure 8b; that is, the relative fluorescence intensity significantly decreased under pretreatment of FA at the folate receptor positive cells, such as Y79 cells and KB cells. Interestingly, blocking of the folate receptors of the cancer cells by FA pretreatment was less effective in the folate receptor negative cells, such as MCF-7 cells (i.e., the decrease in fluorescence intensity was smaller in the MCF-7 cells than those of the folate receptor positive cells). These results indicate that the FAPEGtaCe6 nanophotosensitizers could be intracellularly delivered through a folate receptor mediated pathway. Additionally, H_2_O_2_ was added to the culture medium to investigate the effect of extracellular ROS on the Ce6 uptake ratio. When H_2_O_2_ was added to the FA pretreatment group, the fluorescence intensity of the Y79 cells and KB cells increased, while the fluorescence intensity for the Ce6 treatment was not significantly changed, as shown in Figure 8a,b. These results indicate that the FAPEGtaCe6 nanophotosensitizers could be delivered to the cancer cells via ROS-sensitive mechanisms.

The effect of ROS generation and PDT also supported the results of Figure 8, as shown in Figure 9. Figure 9a,b shows that FA pretreatment did not significantly affect the changes in the intracellular ROS level or PDT-mediated cell death in Ce6 treatment. However, FA pretreatment significantly changed the intracellular ROS level and cell death, as shown in Figure 9a,b (i.e., FA pretreatment significantly decreased the intracellular ROS level and cell viability in nanophotosensitizer treatment). Additionally, the decrease in intracellular ROS level and cell death of folate receptor negative cells, such as MCF-7 cells, was relatively lower than those of folate receptor positive cells. These results clearly indicate that PDT efficacy of FAPEGtaCe6 nanophotosensitizers could also be controlled by folate receptors.

### 2.4. In Vivo Pulmonary Metastasis Model of Y79 Cells for Biodistribution Study

The effect of folate receptor blocking was also assessed with a pulmonary metastasis model of Y79 cells, as shown in Figure 10. For biodistribution of the nanophotosensitizers in mouse organs, the Y79 cells were intravenously (i.v.) administered via the tail veins of the mice. FA (10 mg/kg in 0.1 mL PBS) was i.v. injected to block the folate receptor of the Y79 cells 30 min before injection of the nanophotosensitizers. As shown in Figure 10, the fluorescence intensity without pretreatment of FA (FA, 0 mg/kg) showed a significantly higher fluorescence intensity in the lungs than those of the other organs, indicating that nanophotosensitizers could be efficiently delivered to the pulmonary metastasis of Y79 cells, as shown in Figure 10 (upper images). Otherwise, blocking of the folate receptors of the Y79 cells (FA, 10 mg/kg) in the lungs induced a significant decrease in the fluorescence intensity compared with the other organs. These results indicate that the FAPEGtaCe6 nanophotosensitizers were delivered to the folate receptor positive Y79 cells through a folate receptor-mediated pathway. An in vivo pulmonary metastasis model of the Y79 cells also proved that the FAPEGtaCe6 nanophotosensitizers had targetability against the folate receptors of the tumors.

## 3. Discussion

The tumor microenvironment has a deep relationship with an elevated redox potential compared with normal tissues and cells [31,32,33,34,35]. In the tumor microenvironment, H_2_O_2_ metabolic activity is known to be decreased in tumor cells, and the H_2_O_2_ in the tumor microenvironment is able to be accumulated up to 100 µM, while the hydrogen peroxide in normal cells is normally less than 20 nM [32,33,34]. Furthermore, de Sá Junior et al. found that high ROS levels act as a cancer modulator and induce a genotoxic or proapoptotic effect on cancer cells [34]. They argued that these paradoxical characters of ROS guide antitumor therapeutic approaches using various chemical agents (i.e., molecules such as antioxidant chemicals can be used to prevent ROS formation and then prevent carcinogenesis). Practically, the FAPEGtaCe6 nanophotosensitizers produced intracellular ROS in various cancer cells by light irradiation, and an elevated ROS level induced the death of cancer cells, as shown in Figure 7. However, the viability of the cancer cells was not affected by Ce6 or nanophotosensitizers in the absence of light irradiation (Figure 6). Otherwise, other agents for producing ROS may be used to promote a sudden increase in the ROS and then kill the cancer cells through oxidative stress against the tumor tissues [33,34,35,36]. Low intracellular levels of ROS are required for the signal transduction of normal cells, while high levels of ROS in cancer cells are required to maintain their high proliferation rate and to make the tumor resistant to conventional chemotherapy [35,36]. Therefore, this double-edged sword effect of ROS provides the opportunity to develop therapeutic strategies (i.e., increasing the intracellular ROS concentration until a toxic level can be applied to develop anticancer therapeutic carriers) [35,36,37]. For example, ROS-producing agents such as piperlongumine can be used to induce the apoptotic death of cancer cells via overproduction of intracellular ROS [28,38,39]. Furthermore, photosensitizers are mentioned as a typical therapeutic agent for producing ROS in cancer cells [40]. However, a deficiency of tumor specificity of conventional chemical agents is problematic for application in practical use in humans because these agents also have the potential to elevate ROS levels in normal cells or tissues [40,41]. For example, photosensitizers act on both normal cells and unhealthy cells and make patients sensitive to sunlight for a long duration [42,43]. These drawbacks require a novel delivery system for the specific targeting of cancer with the minimization of photosensitizers against normal cells or tissues. In our results, the FAPEGtaCe6 nanophotosensitizers liberated Ce6 in an ROS-sensitive manner (i.e., the liberation of Ce6 from the nanophotosensitizers was accelerated by the oxidative stress, while Ce6’s release rate was significantly lower in the absence of H_2_O_2_ than in the presence of H_2_O_2_, as shown in Figure 4). These results must have been due to the ROS-specific disintegration of nanophotosensitizers in the presence of H_2_O_2_, as shown in Figure 3. From these points of view, nanocarrier-based drug delivery systems have been investigated and found to be sensitive against higher intracellular ROS levels [27,28,37]. For example, nanofiber mats designed to be sensitive to ROS can be applied to the site-specific release of ROS-producing agents (i.e., anticancer agent release can be accelerated from the nanofiber mats while being sensitive to the ROS level) [43]. In this study, we designed nanophotosensitizers sensitive to ROS (i.e., nanophotosensitizers that responded and disintegrated according to the concentration of H_2_O_2_, a typical ROS molecule) (Figure 3). Furthermore, Ce6’s release from nanophotosensitizers could be accelerated according to the concentration of H_2_O_2_ (i.e., Ce6’s release rate was increased at higher H_2_O_2_ concentrations, as shown in Figure 4). Sun et al. also reported that ROS-sensitive nano-assemblies provide a synergistic anticancer effect on chemotherapy and PDT [44]. They argued that ROS-sensitive nano-assemblies can be used to encourage ROS-mediated PDT and synergistically promote paclitaxel-mediated chemotherapy. In our results, the nanophotosensitizers could be disintegrated in the presence of H_2_O_2_, and Ce6’s release rate from the nanophotosensitizers could also be accelerated as shown in Figure 4. These results indicate that the nanophotosensitizers could be specifically delivered to a disease site having high oxidative stress, and thereafter, liberated Ce6 could be preferentially delivered to the disease cells. Since oxidative stress is normally elevated in cancer cells, these peculiarities of cancer cells can be applied in the targeting of anticancer agents [37,45]. We used higher H_2_O_2_ concentrations than that of the tumor microenvironment for clarity of the ROS sensitivity of the nanophotosensitizers. As shown in Figure 3 and Figure 4, the H_2_O_2_ addition in the nanophotosensitizer solution induced modulation of the particle size distribution. Pandya et al. also reported that paclitaxel-incorporated nanoparticles were degraded by incubation with 5 mM H_2_O_2_, and the particle size was modified [46]. Our group also previously reported that poly(DL-lactide-coglycolide)/poly(ethylene glycol) nanoparticles having diselenide linkages showed H_2_O_2_-dependent degradation, and the antibiotic release rate was dependent on the H_2_O_2_ concentration (i.e., nanoparticle degradation was accelerated at 10 mM H_2_O_2_ rather than 0 or 1 mM H_2_O_2_, and the ciprofloxacin release rate also accelerated) [47]. This study also showed that the Ce6 release rate was significantly increased when 10 mM H_2_O_2_ was added to the release media, as shown in Figure 4. These results indicate that the FAPEGtaCe6 nanophotosensitizers had ROS sensitivity and could respond to the oxidative stress of the tumor microenvironment.

On the other hand, cancer cells are normally characterized as an overexpression of various molecular receptors [48,49]. For example, folate receptors are overexpressed in malignant cells such as Y79 human retinoblastoma cells compared with normal cells such as ARPE-19 cells [50]. This event can also be applied to develop nanomedicine to be sensitive to elevated ROS levels. Alsaab et al. reported that folic acid-decorated nanomicelles represent improved cytotoxicity against retinoblastoma cells, while folate receptor negative normal cells such as ARPE-19 cells are not affected by nanomicelles [51]. In our results, when using non-cancerous normal cell lines such as ARPE-19 and HaKaT cells (Appendix A), the FAPEGtaCe6 nanophotosensitizers showed a higher intracellular Ce6 uptake (Appendix Aa), ROS generation (Appendix Ab) and phototoxicity (Appendix Ac) similar to cancer cells, even though their gap was smaller than those of cancer cells. These results mean that nanophotosensitizers are also able to be delivered to normal cells in a non-specific manner. However, an in vivo study using pulmonary metastasis showed that the nanophotosensitizers were specifically delivered to the Y79 cell-metastasized lungs as shown in Figure 10. These results indicate that FAPEGtaCe6 nanophotosensitizers can be delivered to tumor cells in the biological system of the human body. Folic acid-conjugated graphene oxide significantly increased the accumulation of Ce6 in the tumor cells and led to a remarkable photodynamic efficacy on the MGC803 cells upon irradiation [52]. Feng et al. reported that fluorescent on-off nanoprobes were delivered more efficiently to the folate receptor positive HeLa cells than the folate receptor negative NIH-3T3 cells or MCF-7 cells [53]. Our results also show that the nanophotosensitizers of FAPEGtaCe6 could be delivered via the folate receptors of cancer cells, as shown in Figure 8. Furthermore, ROS production, and thereby the PDT efficacy, of the nanophotosensitizers also showed folate receptor-sensitive behavior, as shown in Figure 9 (i.e., when the folate receptors of the cancer cells were blocked, the ROS production and PDT efficacy significantly decreased). Furthermore, the nanophotosensitizers of FAPEGtaCe6 could be delivered via a folate receptor-mediated pathway in the in vivo pulmonary metastasis model of Y79 cells.

We suggest that FAPEGtaCe6 nanophotosensitizers are a promising candidate for the targeted PDT of cancers having folate receptors.

## 4. Materials and Methods

### 4.1. Materials

The 3-[3-(2-carboxyethoxy)-2,2-bis(2-carboxyethoxymethyl)propoxy]propanoic acid (tetra acid, or TA) and Ce6 were obtained from Frontier Scientific Inc. (Logan, UT, USA). The bifunctional poly(ethylene glycol) (NH_2_-PEG-COOH, molecular weight: 2000 g/mol), folic acid (FA) 3-(4,5-dimethyl-2-thiazolyl)-2,5-diphenyl-2H-tetrazolium bromide (MTT), *N*-(3-dimethylaminopropyl)-*N*′-ethylcarbodiimide hydrochloride (EDAC), *N*-hydroxysuccinimide (NHS), triethylamine (TEA), 2′,7′-dichlorofluorescin diacetate (DCFH-DA), hydrogen peroxide (H_2_O_2_), dimethylsulfoxide (DMSO) and selenocystamine dihydrochloride were purchased from Sigma Chem. Co. (St. Louis, MO, USA). Singlet oxygen sensor green (SOSG) was obtained from Invitrogen (Thermo Fisher Scientific Co. Ltd., Eugene, OR, USA). The dialysis membranes having molecular weight cut-offs (MWCOs) of 1000, 2000 and 8000 g/mol were purchased from Spectra/Por^TM^ Membranes (Spectrum Chem. Mfg. Co., New Brunswick, NJ, USA). Organic solvents were used at the HPLC grade or extra-pure grade.

### 4.2. Synthesis of FAPEGtaCe6 Conjugates

For the FA-PEG-COOH conjugates, 88.2 mg of FA dissolved in 10 mL DMSO was mixed with 38.4 mg EDAC and 23 mg of NHS. This solution was stirred for 3 h to activate the carboxyl group of the FA. Following this, 400 mg of NH_2_-PEG-COOH was added to this solution and then magnetically stirred for 24 h. This was introduced into the dialysis membrane (MWCO, 1000 g/mol) and then dialyzed against 1 L of distilled water for 1 day. Distilled water was exchanged in 2–3-h intervals to remove organic solvents. The dialyzed solution was lyophilized for 2 days. The yield of the lyophilized solid was calculated as follows: yield = [weight of lyophilized solid/(weight of FA + weight of PEG)] × 100. The yield of lyophilized solid was higher than 97 % (*w*/*w*).

For the FA-PEG-sese conjugates, 242 mg of FA-PEG-COOH dissolved in 10 mL DMSO was mixed with 19.2 mg EDAC and 11.5 mg NHS. This solution was stirred for 3 h to activate the carboxylic end group of FA-PEG-COOH, and after that, more than 5 equivalents of selenocystamine HCl were added with TEA. This solution was magnetically stirred for 1 day. After that, this solution was dialyzed against 1 L of distilled water for 2 days. Distilled water was exchanged in 3-h intervals to remove organic solvents. The dialyzed solution was lyophilized for 2 days.

Regarding the TA-sese-Ce6 conjugates, to synthesize the Ce6-sese conjugates, 59.7 mg of Ce6 dissolved in 10 mL DMSO was mixed with 19.2 mg EDAC and 11.5 mg NHS. This solution was stirred for 3 h to activate the carboxylic end group of Ce6. After that, 32 mg of selenocystamine HCl was dissolved in 5 mL of a DMSO/water mixture (DMSO/water = 4/1 *v*/*v*). This solution was magnetically stirred for 9 h. For activation of the carboxylic acid of TA, 10.6 mg TA dissolved in 5 mL DMSO was mixed with 19.2 mg EDAC and 11.5 mg NHS. This solution was magnetically stirred for 6 h. After that, this solution was mixed with a solution of Ce6-sese conjugates and then magnetically stirred for 24 h. Following this, the resulting solution was introduced into the dialysis membrane (MWCO: 2000 g/mol). This was dialyzed against deionized water for 2 days with an exchange of water in 2–3-h intervals to avoid saturation and to remove unreacted chemicals. The resulting solution was lyophilized for 3 days, and the obtained solid was used for the next synthesis step or stored in a refrigerator (4 *°*C). The yield of the TA-sese-Ce6 conjugates was calculated as follows: yield = [weight of TA-sese-Ce6/(weight of Ce6-sese + weight of TA)] × 100. The yield of the TA-sese-Ce6 conjugates was higher than 95% (*w*/*w*).

For synthesis of the FAPEGtaCe6 conjugates, 91 mg of the TA-sese-Ce6 conjugates was dissolved in 20 mL DMSO and mixed with 4 equivalents of an EDAC/NHS system. This solution was stirred for 6 h to activate the carboxyl group of the TA-sese-Ce6. Following this, 265 mg of the FA-PEG-sese conjugates was added to this solution and then magnetically stirred for 2 days. This solution was introduced into the dialysis membrane (MWCO: 8000 g/mol) and then dialyzed against deionized water for 2 days with an exchange of water in 2–3-h intervals. The resulting solution was lyophilized for 3 days, and a dark green solid was obtained. The yield of the FAPEGtaCe6 conjugates was calculated as follows: yield = [weight of FAPEGtaCe6 conjugates/(weight of TA-sese-Ce6 + weight of FA-PEG-sese)] × 100. The yield of the FAPEGtaCe6 conjugates was higher than 94% (*w*/*w*).

### 4.3. Characterization of FAPEGtaCe6 Conjugates

^1^H nuclear magnetic resonance (NMR) spectroscopy (500 mHz superconducting Fourier transform (FT)-NMR spectrometer, Varian Unity Inova 500 MHz NB High Resolution FT NMR; Varian Inc, Santa Clara, CA, USA) was used to monitor the synthesis procedure.

### 4.4. Fabrication of Nanophotosensitizers

The FAPEGtaCe6 conjugates (20 mg) were dissolved in 5-mL water/DMSO mixtures (1/4, *v*/*v*) and then introduced into the dialysis membrane (MWCO: 8000 g/mol) as reported previously [30,54]. This solution was dialyzed for 1 day with an exchange of water every 2–3 h to remove the organic solvent. The resulting solution was used for analysis or a PDT effect.

### 4.5. Ce6 Contents in the FAPEGtaCe6 Nanophotosensitizers

The contents of Ce6 in the conjugates were estimated as follows. The FAPEGtaCe6 nanophotosensitizers were fabricated as described above and were adjusted to 1 mg/mL. Then, 5 mL of this solution was mixed with 45 mL of phosphate-buffered saline (PBS, 0.01 M, pH 7.4) in the presence of 20 mM H_2_O_2_. This solution was stirred for 48 h and then diluted with DMSO more than 10 times. The Ce6 concentration was measured with an Infinite M200 pro microplate reader (Tecan, Männedorf, Switzerland) (excitation wavelength: 407, emission wavelength: 664 nm). The Ce6 itself dissolved in DMSO and was diluted 20 times with PBS (20 mM H_2_O_2_). This was similarly used instead of a nanophotosensitizer solution for the standard test. Ce6 content (wt.%) = (Ce6 weight/total weight of nanophotosensitizers) × 100, where the Ce6 contents were about 16.2% (*w*/*w*).

### 4.6. Characterization of Nanophotosensitizers

The particle size of the FAPEGtaCe6 nanophotosensitizers was measured with Zetasizer Nano-ZS (Malvern, Worcestershire, UK). For measurement, the concentration of the nanophotosensitizer solution was adjusted to 0.1 % (*w*/*w*).

A transmission electron microscope (TEM) (H-7600, Hitachi Instruments Ltd., Tokyo, Japan) was employed to observe the morphology of the nanophotosensitizers. One drop of the nanophotosensitizer aqueous solution was dropped onto the carbon film-coated grid for the TEM. This was dried at room temperature for 3 h. Following this, phosphotungstic acid (0.1%, *w*/*w* in deionized water) was used to stain the nanophotosensitizers negatively. The observation was carried out at 80 kV.

The UV absorption spectra of the Ce6 or FAPEGtaCe6 nanophotosensitizers were measured with a Genesys 10s UV-VIS spectrophotometer (Thermo Fisher Scientific, Waltham, MA, USA).

### 4.7. Fluorescence Spectra

A fluorescence emission scan was measured between 500 nm and 800 nm (excitation wavelength: 400 nm) with a multifunctional microplate reader (Infinite M200 pro microplate reader, Tecan, Mannedorf, Switzerland). A similar solution was fluorescently observed with a Maestro 2 small animal imaging instrument (Cambridge Research and Instrumentation Inc., Woburn, MA 01801, USA). The nanophotosensitizers in the PBS was mixed with various concentrations of hydrogen peroxide and then incubated for 4 h at 37 °C.

### 4.8. Singlet Oxygen (SO) Generation of Nanophotosensitizers

SO generation from the Ce6 or nanophotosensitizers in an aqueous solution was measured using 1 mL of Ce6 alone or a nanophotosensitizer solution (5 μg/mL Ce6 equivalent in distilled water, 1% DMSO) [55]. For this solution, an SOSG reagent was added (final concentration: 5 µM). Each solution was irradiated with an expanded homogenous beam (664 nm, SH Systems, Gwangju, Korea) at different time points (0.5, 1, 2, 5, and 10 min). After that, the fluorescence intensity of this solution was measured with a fluorescence spectrophotometer (RF-5301PC, Shimadzu Co., Kyoto, Japan) at a 488-nm excitation wavelength and 525-nm emission wavelength. This procedure was carried out under dark conditions.

### 4.9. Drug Release Study

A Ce6 release study was carried out as follows. First, 5 mg of nanophotosensitizers (5 mg/5 mL deionized water), fabricated as described above, was introduced into a dialysis tube (MWCO: 8000 g/mol). Then, this was put into a 50-mL conical tube with 45 mL PBS (0.01 M, pH 7.4). Hydrogen peroxide was added 3 h later to study the effect of hydrogen peroxide. This was incubated in 37 °C with shaking at 100 rpm. The whole media was taken to evaluate the Ce6 release rate at predetermined time intervals and then replaced with fresh PBS. The Ce6 concentration in the media was estimated with an Infinite M200 pro microplate reader (Tecan) (excitation wavelength: 407 nm; emission wavelength: 664 nm). All experiments were performed in triplicate, and the results were expressed as the mean ± standard deviation (S.D.).

### 4.10. Cell Culture

Y79 retinoblastoma cells and ARPE-19 human retinal pigment epithelial cells were purchased from the American Type Culture Collection (ATCC, Manassas, VA, USA). HaCaT human keratinocyte cells as well as KB human epithelial and MCF-7 human breast cancer cells were obtained from the Korean Cell Line Bank (Seoul, Korea). Y79, MCF-7 and KB cells were maintained with RPMI1640 medium (Gibco, Grand Island, NY, USA) at 37 °C in a 5% CO_2_ incubator. The HaCaT cells were maintained in DMEM medium (Gibco, Grand Island, NY, USA) at 37 °C in a 5% CO_2_ incubator. The ARPE-19 cells were maintained using DMEM/F12 medium (Gibco, Grand Island, NY, USA) at 37 *°*C in a 5% CO_2_ incubator. All media for the cell culture were supplemented with 10% heat-inactivated fetal bovine serum (FBS) (Invitrogen) and 1% penicillin/streptomycin.

### 4.11. PDT Treatment

The Y79, MCF-7 and KB cells (2 × 10^4^ cells/well) were seeded in 96 well plates and then exposed to Ce6 or nanophotosensitizers. For the Ce6 treatment, it was dissolved in DMSO and then diluted with serum-free media more than 100 times. The nanophotosensitizers fabricated as described above were diluted with serum-free media. Each treatment was incubated for 2 h in a 5% CO_2_ incubator at 37 °C. After that, the cells were washed with PBS twice, replaced with 100 µL serum-free media and adopted for PDT treatment. For PDT treatment, the cells were irradiated at 664 nm using an expanded homogenous beam (SH Systems, Gwangju, Korea) at 2.0 J/cm^2^ (the signal was measured using a photo-radiometer (DeltaOhm, Padova, Italy)). The light intensity of the expanded homogenous beam and irradiation time were 3.515 mW/cm^2^ and 569 s (2000 mJ/cm^2^), respectively. After that, the cells were incubated for 24 h in a CO_2_ incubator at 37 *°*C. The viability of the cells was evaluated with an MTT proliferation assay, and 30 μL of an MTT solution (5 mg/mL in PBS) was added to the wells and further incubated for 3 h in a CO_2_ incubator. The supernatants were discarded and replaced with 100 µL DMSO. The absorbance (570 nm) was measured with an Infinite M200 pro microplate reader. All of the procedure was performed in dark conditions.

For dark toxicity, the ARPE-19, Y79, MCF-7 and KB cells (2 × 10^4^ cells/well) were seeded in 96 well plates and then similarly treated as described above without light irradiation.

### 4.12. Intracellular Uptake of Ce6 or Nanophotosensitizers

The cancer cells (2 × 10^4^ cells) were seeded in a 96-well plate and then exposed to Ce6 or nanophotosensitizers for 2 h as described above. The cells were washed with PBS twice and solubilized in 50 µL of lysis buffer (GenDEPOT, Barker, TX, USA). Then, the Ce6 uptake ratio was measured with an Infinite M200 pro microplate reader (Tecan) (excitation wavelength: 407 nm; emission wavelength: 664 nm).

For fluorescence observation, 3 × 10^5^ cells seeded in 6 well plates with cover glass were exposed to Ce6 or nanophotosenstitizers for 90 min. The cells were washed with PBS twice, fixed with a 4% paraformaldehyde solution and then immobilized with a mounting solution (Immunomount, thermo Electron Co. Pittsburgh, PA, USA). Observation of the cells was performed with a fluorescence microscope (Eclipes 80i; Nikon, Tokyo, Japan).

### 4.13. ROS Assay

The cells (2 × 10^4^ cells) were seeded in a 96-well plate and then exposed to Ce6 or nanophotosensitizers for 2 h as described above. To measure the generation of intracellular ROS, a DCFH-DA assay was employed. The cells were exposed to Ce6 or HAssCe6 nanophotosensitizers in phenol red free RPMI media in the presence of DCFH-DA (final concentration: 20 µM) for 2 h at 37 °C. Following this, the cells were washed with PBS twice, replaced with 100 µL fresh phenol red free RPMI media and then irradiated at 664 nm (2.0 J/cm^2^). Intracellular ROS generation was estimated with a microplate reader (Infinite M200 pro microplate reader (Tecan); excitation wavelength: 485 nm; emission wavelength: 535 nm).

### 4.14. In Vivo Animal Tumor Imaging

A pulmonary metastasis model using nude BALb/C mice (male, 20 g, 5 weeks old) was prepared using the Y79 cells for in vivo fluorescence imaging. The Y79 cells (5 × 10^5^ cells/0.1 mL PBS) were intravenously (i.v.) administered via the tail veins of the nude BALb/C mice. Three weeks later, nanophotosensitizers (10 mg/kg) were sterilized with a 0.8 µm syringe filter and i.v. administered via the tail veins of nude BALb/C mice. The injection volume was 0.1 mL, and 30 min before injection of the nanophotosensitizers, FA (10 mg/kg, in 0.1 mL PBS) was i.v. injected to block the folate receptor of the Y79 tumor. One day later, the mice were sacrificed and dissected to observe the organs and to observe the biodistribution of the nanophotosensitizers with a Maestro^TM^ 2 small animal imaging instrument. For fluorescence observation of each organ, the filter of the Maestro^TM^ 2 small animal imaging instrument was an orange filter set (excitation filter: 586-631 nm; emission filter: 645 nm longpass).

### 4.15. Statistical Analysis

A Student’s *t* test using SigmaPlot^®^ (SigmaPlot^®^ v.11.0, Systat Software, Inc., San Jose, CA, USA) was used to estimate the statistical significance of the results and evaluate *p* < 0.05 as the minimal level of significance.

## 5. Conclusions

The FAPEGtaCe6 conjugates were synthesized for the ROS-specific and folate receptor-specific delivery of photosensitizers. The nanophotosensitizers were fabricated by a dialysis procedure. The nanophotosensitizers of the FAPEGtaCe6 conjugates showed spherical shapes and had small particle sizes of less than 200 nm. The nanophotosensitizers showed sensitivity (i.e., they disintegrated with the addition of H_2_O_2_, and the particle size distribution was changed from a monomodal pattern to a multimodal pattern). Furthermore, the addition of H_2_O_2_ induced an increase in the fluorescence intensity and Ce6 release rate from the nanophotosensitizers, indicating that the nanophotosensitizers had ROS-sensitive properties and the Ce6 release could be controlled by the ROS. The Ce6 uptake ratio, ROS generation and phototoxicity of the nanophotosensitizers were significantly higher than those of the free Ce6 against various cancer cells in vitro. In particular, the nanophotosensitizers showed a folate receptor-mediated delivery capacity against folate receptor-overexpressing Y79 cells and KB cells (i.e., blocking by the folate receptor via pretreatment with an FA-induced decrease in the intracellular Ce6 uptake, decreased ROS generation and, thereby, photoxicity). These results indicate that the nanophotosensitizers of the FAPEGtaCe6 conjugates had folate receptor specificity in vitro. Furthermore, the pulmonary metastasis model using Y79 cells showed a folate receptor-specific delivery capacity for the nanophotosensitizers. When the folate receptor of the Y79 cells in the pulmonary metastatic model was blocked, delivery of the nanophotosensitizers to the lungs was inhibited, indicating that the nanophotosensitizers of the FAPEGtaCe6 conjugates had folate receptor specificity in vitro and in vivo. We suggest that the nanophotosensitizers of FAPEGtaCe6 conjugates are promising candidates for a targeted PDT approach against cancer cells.

## Figures and Tables

**Figure 1 ijms-23-03117-f001:**
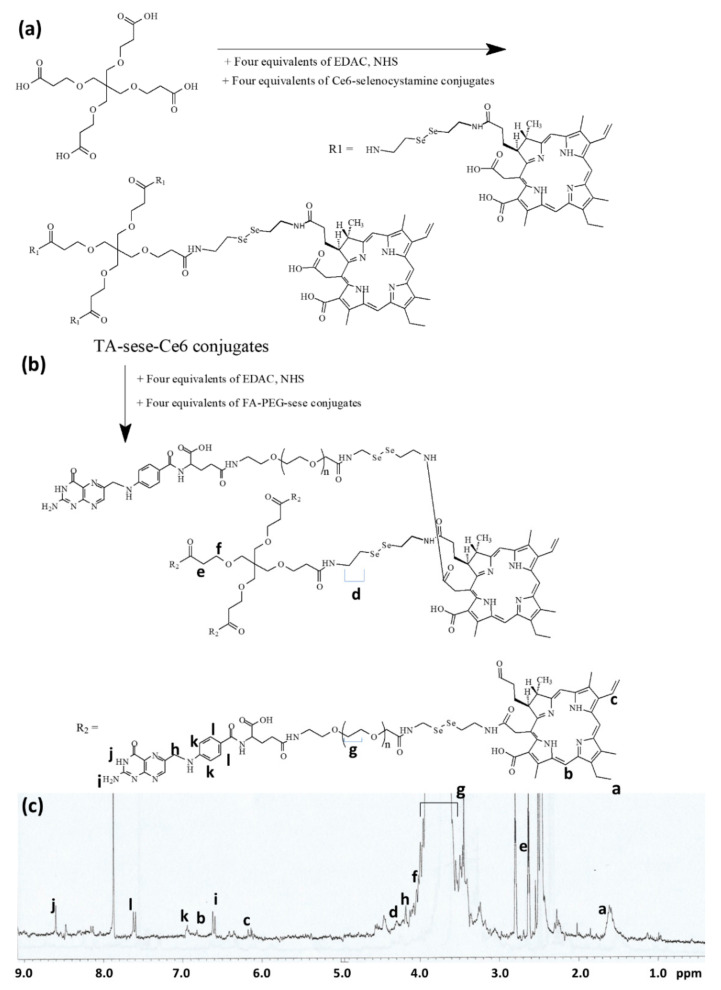
Synthesis scheme and ^1^H NMR spectra of FAPEGtaCe6 conjugates. Synthesis scheme of TA-sese-Ce6 tetramer (**a**) and FAPEGtaCe6 conjugates (**b**). (**c**) ^1^H NMR spectra of FAPEGtaCe6 conjugates.

**Figure 2 ijms-23-03117-f002:**
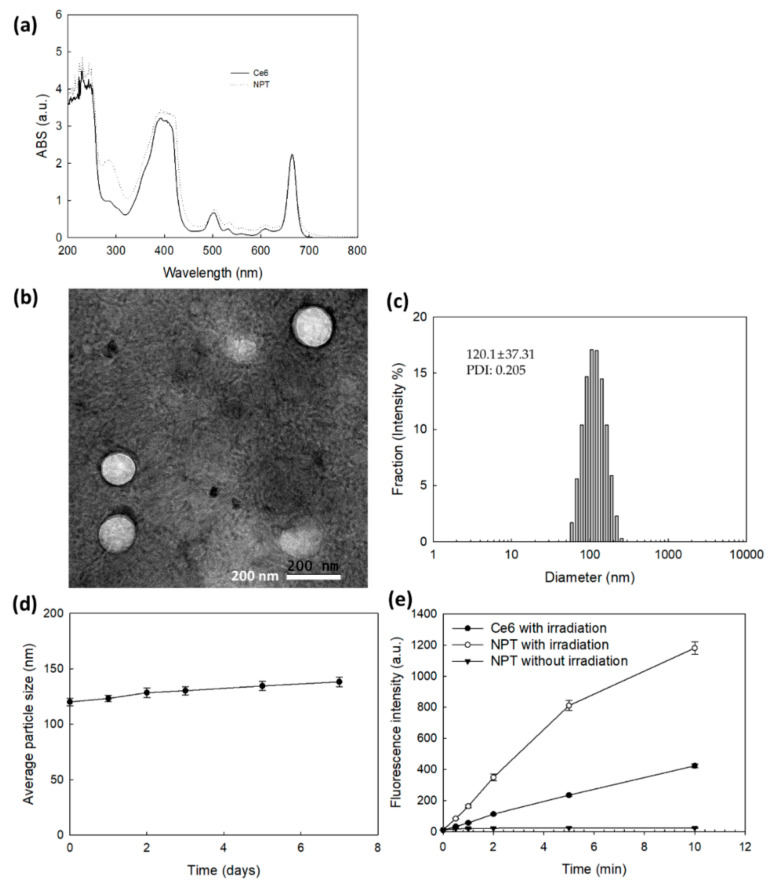
(**a**) UV-VIS absorption of Ce6 or FAPEGtaCe6 nanophotosensitizers in DMSO (0.1 mg/mL Ce6 equivalent). (**b**) TEM image and (**c**) particle size distribution of FAPEGtaCe6 nanophotosensitizers. (**d**) Stability of FAPEGtaCe6 nanophotosensitizers. For the stability test, aqueous nanophotosensitizer solution (1 mg FAPEGtaCe6 nanophotosensitizers/mL deionized water) was left at room temperature (20 °C). (**e**) Singlet oxygen generation from Ce6 alone or FAPEGtaCe6 nanophotosensitizers under light irradiation at 664 nm (*n* = 4).

**Figure 3 ijms-23-03117-f003:**
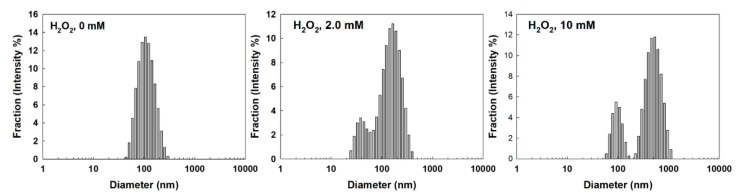
The effect of H_2_O_2_ concentration on the changes in particle size distribution. To study ROS sensitivity, nanophotosensitizers in PBS (1 mg/mL) were incubated in the presence of H_2_O_2_ at 37 °C for 3 h.

**Figure 4 ijms-23-03117-f004:**
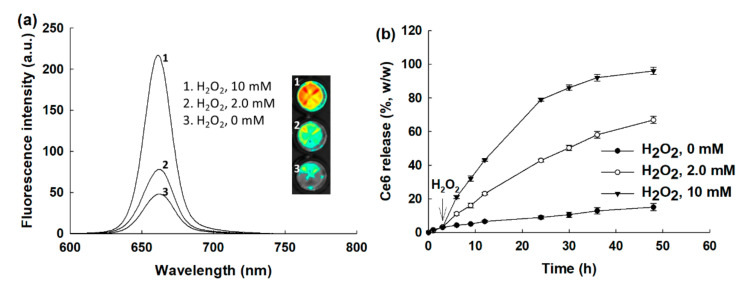
The effect of H_2_O_2_ concentration on the changes in the fluorescence emission spectra (**a**) and drug release (**b**). To measure fluorescence emission spectra, nanophotosensitizers in PBS (0.1 mg/mL) were incubated in the presence of H_2_O_2_ at 37 °C for 3 h.

**Figure 5 ijms-23-03117-f005:**
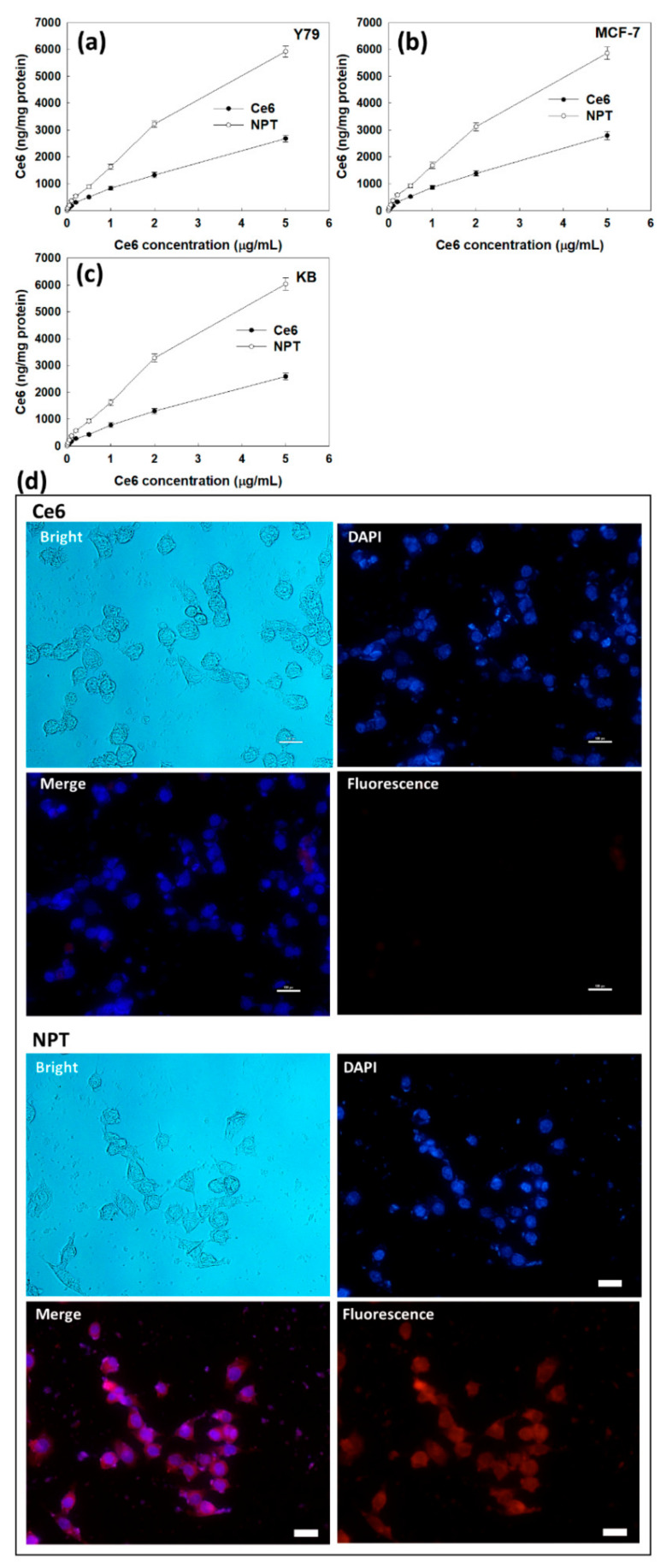
Ce6 uptake of Y79 cells (**a**), MCF-7 cells (**b**) and KB cells (**c**). Fluorescence observation of KB cells with treatment of Ce6 or nanophotosensitizers (**d**). Bar = 20 µm.

**Figure 6 ijms-23-03117-f006:**
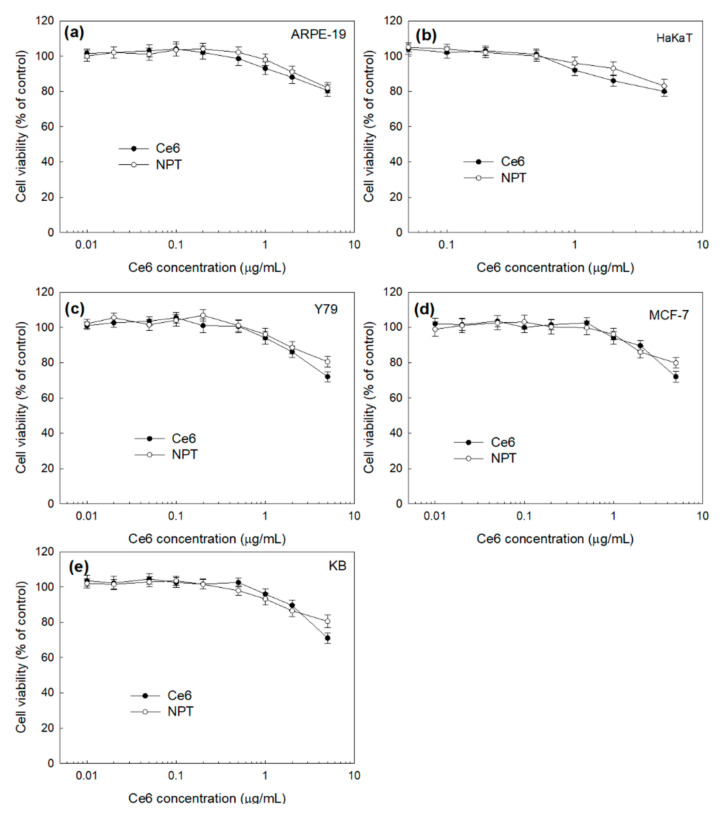
Dark toxicity of free Ce6 and nanophotosensitizers, with normal cell lines such as ARPE-19 cells (**a**) and HaKaT cells (**b**) and cancer cell lines such as Y79 cells (**c**), MCF-7 cells (**d**) and KB cells (**e**). NPT = FAPEGtaCe6 nanophotosensitizers.

**Figure 7 ijms-23-03117-f007:**
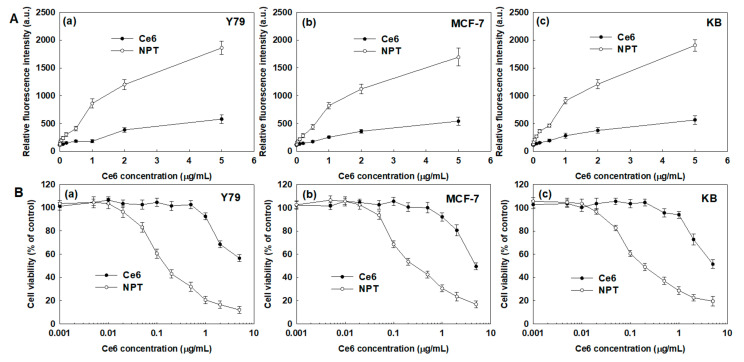
The effect of free Ce6 or nanophotosensitizers on ROS generation (**A**) and phototoxicity (**B**) against various cancer cells. (**A**). (**a**) Y79 cells, (**b**) MCF-7 cells and (**c**) KB cells. (**B**). (**a**) Y79 cells, (**b**) MCF-7 cells and (**c**) KB cells. DCFH-DA assay was employed to measure intracellular ROS level. Cells were irradiated at 664 nm (2 J/cm^2^). All values are average ± S.D. from results of a single independent experiment with eight replicates.

**Figure 8 ijms-23-03117-f008:**
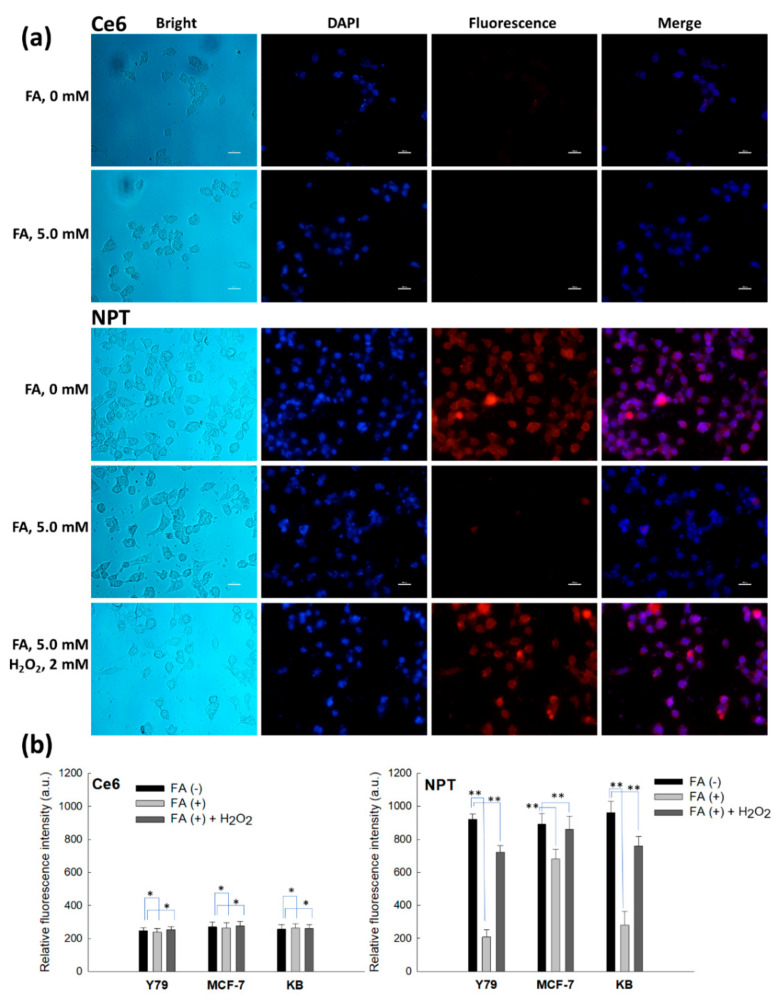
Fluorescence observations of KB cells (**a**) and intracellular Ce6 uptake against various cancer cells (**b**). FA (5.0 mM) was pretreated to the KB cells 30 min before treatment of Ce6 or nanophotosensitizers. For H_2_O_2_ (2 mM) treatment, Ce6 or nanophotosensitizers were treated to cells, and then H_2_O_2_ was added to the medium. Ce6 concentration was 2 µg/mL. Bar = 20 µm. *, ** *p* < 0.01.

**Figure 9 ijms-23-03117-f009:**
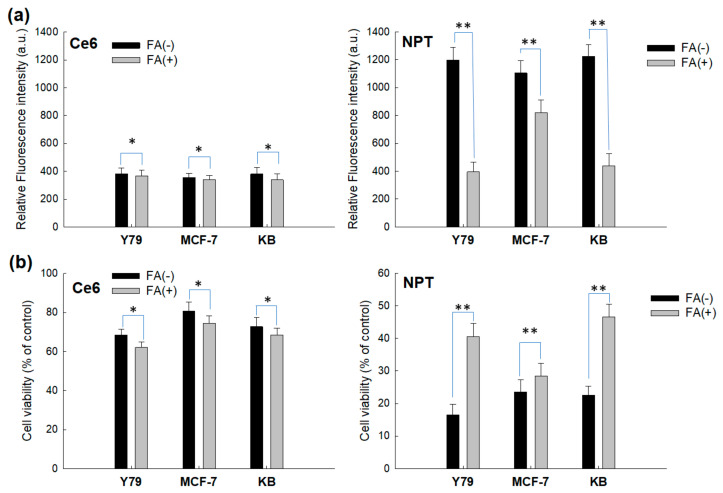
The effect of pretreatment of FA on the intracellular ROS generation (**a**) and phototoxicity (**b**). Cells were treated with Ce6 or nanophotosensitizers in a similar manner to Figure 8. FA (5 mM) was pretreated to the cells 30 min before free Ce6 or nanophotosensitizer treatment. Ce6 concentration was 2 µg/mL. Cells were exposed to free Ce6 or nanophotosensitizers for 2 h and then irradiated at 664 nm (2 J/cm^2^). *, ** *p* < 0.01.

**Figure 10 ijms-23-03117-f010:**
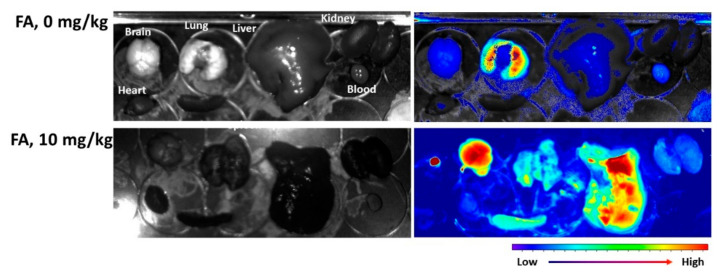
Pulmonary metastasis model of Y79 cells for evaluation of targetability of FAPEGtaCe6 nanophotosensitizers. For pulmonary metastasis of Y79 cells, 5 × 10^5^ cells/0.1 mL PBS were intravenously (i.v.) administered via the tail veins of nude BALb/C mice. Three weeks later, nanophotosensitizers (10 mg/kg) were administered via the tail veins of nude BALb/C mice. For blocking of the folate receptors of Y79 cells in the mouse lungs, FA (10 mg/kg, in 0.1 mL PBS) was i.v. injected 30 min before injection of nanophotosensitizers. One day later, the mice were sacrificed and dissected to observe the organs and the biodistribution of nanophotosensitizers with a Maestro^TM^ 2 small animal imaging instrument.

**Table 1 ijms-23-03117-t001:** Characterization of FAPEGtaCe6 nanophotosensitizers.

	Ce6 Content (*w*/*w*)		Particle Size (nm) ^b^
	Theoretical ^a^	Experimental ^a^	
TA-sese-Ce6 FAPEGtaCe6	65.4 16.8	64.9 16.2	- ^c^ 120.1 ±37.31

^a^ Theoretical contents of Ce6 were calculated from molecular structures and compositions of each molecule of FAPEGtaCe6 nanophotosensitizers. For experimental contents of Ce6, nanophotosensitizers were treated with H_2_O_2_, diluted with DMSO and then measured spectrophotometrically. Ce6 contents were calculated with the following equation: Ce6 content (wt.%) = (Ce6 weight/total weight of nanophotosensitizers) × 100. ^b^ Particle size distributions are shown in Figure 2b. ^c^ Particle size of TA-sese-Ce6 is not determined.

## Data Availability

Not applicable.

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
