# Peer review of "Reactive Oxygen Species and Folate Receptor-Targeted Nanophotosensitizers Composed of Folic Acid-Conjugated and Poly(ethylene glycol)-Chlorin e6 Tetramer Having Diselenide Linkages for Targeted Photodynamic Treatment of Cancer Cells"

_ijms, 2022, doi:10.3390/ijms23063117_

Round 1
Reviewer 1 Report
The article entitled "Reactive oxygen species and folate receptor-targeted nanophotosensitizers composed of folic acid-conjugated and poly(ethylene glycol)-chlorin e6 tetramer having diselenide linkages for targeted photodynamic treatment of cancer cells" deals with the preparation of nanophotosensitizers base on folic acid, PEG, chlorin e6 and some unspecified tetra acid.
The entire work is written in a very confusing way and raises several doubts, not purely substantive, but above all regarding the work methodology.
The two main drawbacks, which greatly influence the rating of the article are:
(1) The authors do not mention the self-organization of the described systems in any way; in this case, it seems that it will strongly depend on the concentration. What is the stability of this material over time? What is the stability when changing the concentration? The authors boldly call their material "nanophotosensitizers." However, there is no word in the introduction, methodology, or discussion of what kind of nanomaterials they are.
(2) The authors seem to use mental shortcuts a lot. It is unknown what their relationship is made of (there is no such thing as tetra acid); it is also unknown where the in vitro tests were ultimately carried out - abstract, the methodology, and the results have different records.
Specific comments:
Title: too long and confusing.
Abstract: It is not constructed properly. It should refer to the purpose of the work, description of the methodology, results and summary. Now, it contains unexplained abbreviations, grammar, editorial errors, repetitions and typos. Eg. Conjugates were conjugated again with conjugates to make conjugates(see line 19), “nanophotosensitizers showed sensivity i.e. they disintegrated by addition H2O2 was added and the particle size distribution was changed” (line 24), “phototoxicity” line 29 vs photoxicity” line 33. Etc.
Introduction: The Authors refer to series of polymeric nanocarriers and their superiority in the area of drug delivery, but the references lead to “bulk” polymeric nanocarriers, where in fact they deal with self-organizing compounds.
Results: Title in Table 1 is not adequate to the content of the Table, and actually the entire article. The Scheme and NMR spectra presented in Figure 1 are hardly readable. The Authors state, that specific peaks are presented in the range 1-10 ppm; most of organic signals appear there, so that’s not a confirmation (not as such statement).
Authors refer to some data, which they do not show (e.g. line 116). How was the experimental and theoretical content of Ce6 established? What do you mean by dialysis procedure? Is it commonly used? Is there any reference behind it? In line 144 the Authors refer to Fig 1a, I guess they mean Fig 2a. Line 146, the Authors say aqueous solution, but in methods they describe sample preparation id DMSO/H2O mixture. TEM images are very dark, in fact this image should be repeated. Line 152 lack of unit when the size is given. Moreover, the Authors state the material is 117.1+-45.18 nm big. How did the Authors get that value? Figure 3-10 are to small and in fact unreadable. Not all description contains full explanation of the figures, e,g. in Fig 3 no explanations to a-c is given. Authors use shortcuts which in general (but not all) are explained further in the text. They should be explained at their first appearance. How does fig 5a-c explain the cell viability? What time was the MTT test performed? Line 212-13 – entire sentence is not clear. What do Authors mean by that? Which cell lines the Authors actually took to what test? It is not clear, not even after Materials and Methods explanation.
Discussion: Discussion is very poor in regards to the amount of results obtained. Especially when taking into consideration, that lines 285 (beginning) to 312 are general literature statements, not discussion.
Materials and methods: Once again, what is tetra acid? Sentence starting in line 352 is grammarly incorrect. Line 357; not to activate carboxylic acid, but carboxylic groups. Yield presented by the Authors is not actual yield. Many times the time of the reaction is not specified. What do Authors mean by “more than 24 h”? 25? 48? 200? If the reaction needs 24 h, you should state “at least 24h” etc.
Line 408 – this should be a new subsection. Line 411: diluted more than 10 times? Section 4.8. no explanation of cell lines is given. Sections 4.9, 4.10 and 4.11 which cell lines?
Conclusions: needs to be rewritten.
To summarize, the Authors need to carefully read and improve their article before it can be accepted.
Author Response
Reponse to Reviewer 1’s comment
The article entitled "Reactive oxygen species and folate receptor-targeted nanophotosensitizers composed of folic acid-conjugated and poly(ethylene glycol)-chlorin e6 tetramer having diselenide linkages for targeted photodynamic treatment of cancer cells" deals with the preparation of nanophotosensitizers base on folic acid, PEG, chlorin e6 and some unspecified tetra acid.
The entire work is written in a very confusing way and raises several doubts, not purely substantive, but above all regarding the work methodology.
The two main drawbacks, which greatly influence the rating of the article are:
(1) The authors do not mention the self-organization of the described systems in any way; in this case, it seems that it will strongly depend on the concentration. What is the stability of this material over time? What is the stability when changing the concentration? The authors boldly call their material "nanophotosensitizers." However, there is no word in the introduction, methodology, or discussion of what kind of nanomaterials they are.
Answer) Thank you for your valuable comment. In this report, we used the term “nanophotosensitizers”, which means photosensitizer-incorporated nanoparticles. According to your comment, we mentioned this term in the introduction part. In stability issues, FAPEGtaCe6 nanoparticles maintained its stability, i.e. mean particle sizes were not significantly changed and did not aggregate in the aqueous solution. We indicated it in the Results section.
In introduction section
Especially, nanophotosensitizers, which mean photosensitizer-incoporated nano-sized carrier, have been suggested as an ideal candidate for PDT [12-14]. Various biocompatible polymers or nanomaterials such as natural polysaccharide, dendrimers, iron oxide nanoparticles and synthetic polymers have been investigated to fabricate nanophotosensitizers [12-16]. Chin et al., reported that poly(vinylpyrrolidone) (PVP)-Ce6 formulations accelerate tumor accumulation of photosensitizers and represent fast clearance from skin of nude mice [11]. Poly-cationic polysaccharide such as chitosan can be used to fabricate Ce6-encapsulated nanophotosensitizers [12]. They argued that nanophotosensitizers can be formed by ion complex formation between water-soluble chitosan and Ce6 (ChitoCe6). ChitoCe6 nanophotosensitizers showed superior accumulation in the tumor tissue of nude mice compared to free Ce6 and then improve PDT efficacy against gastrointestinal cancer cells [12].
In our previous report, hyaluronic acid (HA) as a hydrophilic segment and hyperbranched Ce6 as a lipophilic segment formed nano-dimensional carriers as nanophotosensitizers through self-assembling process, i.e. hyperbranched Ce6 formed inner -core of the nanophotosensitizers and HA exposed on the surface of nanophotosensitizers [30]. They argued that nanophotosensitizers composed of HA-hyperbranched Ce6 conjugates via disulfide linkages represent CD44-specific and redox-sensitive delivery capacity of Ce6 capacity against tumor [30]. Those nanophotosensitizers showed CD44-specific imaging and PDT efficacy against U87MG glioblastoma cells. Sun et al., also reported that folic acid-decorated nanocarriers show folate-receptor mediated targeting and then permit MRI-guided imaging of tumor [31].
Furthermore, FAPEGtaCe6 nanophotosensitizers maintained their aqueous stability at least for one week, i.e. mean particle sizes were not significantly changed for 7 days even though their average particle sizes were slightly increased as shown in Figure 2(c). Furthermore, large aggregates and/or precipitants were not observed even after 7 days later. These results indicated that FAPEGtaCe6 nanophotosensitizers can be stored as a aqueous solution for biological application.
Figure 2. TEM image (a) and particle size distribution (b) of FAPEGtaCe6 nanophotosensitizers. (c) Stability of FAPEGtaCe6 nanophotosensitizers. For stability test, aqueous nanophotosensitizer solution (1 mg FAPEGtaCe6 nanophotosensitizers/ml deionized water) was left in the room temperature (20oC).
(2) The authors seem to use mental shortcuts a lot. It is unknown what their relationship is made of (there is no such thing as tetra acid); it is also unknown where the in vitro tests were ultimately carried out - abstract, the methodology, and the results have different records.
Answer) Thank you for your valuable comment. In fact, we purchased tetra-acid from Frontier Scientific (https://www.frontiersci.com/product/tetraacid/). As you indicated, tetra acid is not exact chemical name and therefore its chemical name was indicated in the Introduction/experimental section.
In this study, we synthesized folic acid-conjugated poly(ethylene glycol)/Ce6 tetramer (Abbreviated as FAPEGtaCe6) conjugates using 3-[3-(2-carboxyethoxy)-2,2-bis(2-carboxyethoxymethyl)propoxy]propanoic acid (tetra acid, TA) and fabricated nanophotosensitizers. Tetra acid (TA) was used to synthesize Ce6 tetramer via diselenide linkages since diselenide linkages can be disintegrated by ROS [26,27]. Their targeting efficiency against folate receptor and ROS-sensitivity against cancer cells were studied using Y79 retinoblastoma cells, KB epithelial carcinoma cells and MCF-7 human breast cancer cells. We characterized physicochemical and biological properties of FAPEGtaCe6 nanophotosensitizers in vitro and in vivo.
4.1. Materials
3-[3-(2-carboxyethoxy)-2,2-bis(2-carboxyethoxymethyl)propoxy]propanoic acid (tetra acid, TA)
Specific comments:
Title: too long and confusing.
Answer) Thank you for your valuable comment. According to your comment, we changed the title of our manuscript
Reactive oxygen species and folate receptor-targeted nanophotosensitizers for targeted photodynamic treatment of cancer cells
Abstract: It is not constructed properly. It should refer to the purpose of the work, description of the methodology, results and summary. Now, it contains unexplained abbreviations, grammar, editorial errors, repetitions and typos. Eg. Conjugates were conjugated again with conjugates to make conjugates(see line 19), “nanophotosensitizers showed sensivity i.e. they disintegrated by addition H2O2 was added and the particle size distribution was changed” (line 24), “phototoxicity” line 29 vs photoxicity” line 33. Etc.
Answer) Thank you for your valuable comment. According to your comment, we checked all of the manuscript and revised them.
Abstract: Folic acid-conjugated nanophotosensitizers composed of folic acid (FA), poly(ethylene glycol) (PEG) and chlorin e6 (Ce6) tetramer was synthesized using diselenide linkages for ROS-specific and folate receptor-specific delivery of photosensitizers. Ce6 was conjugated with 3-[3-(2-carboxyethoxy)-2,2-bis(2-carboxyethoxymethyl)propoxy]propanoic acid (tetra acid, TA) to make Ce6 tetramer via selenocystamine linkages (TA-sese-Ce6 conjugates). In the carboxylic acid end group of TA-sese-Ce6 conjugates, FA-PEG was attached again using selenocystamine linkages to make FA-PEG/TA-sese-Ce6 conjugates (Abbreviated as FAPEGtaCe6 conjugates). Nanophotosensitizers were fabricated by dialysis procedure. In morphological observations, they showed spherical shapes with small diameter less than 200 nm. Stability of aqueous FAPEGtaCe6 nanophotosenstizer solution was maintained, i.e. their particle sizes were not significantly changed until 7 days. When H2O2 was added to nanophotosensitizer solution, particle size distribution was changed from monomodal pattern to multimodal pattern. Also, fluorescence intensity and Ce6 release rate from nanophotosensitizers were also increased by addition of H2O2. These results indicated that nanophotosensitizers have ROS-sensitive properties. In vitro cell culture study, FAPEGtaCe6 nanophotosenstizer treatment against cancer cells increased Ce6 uptake ratio, ROS generation and light-irradiated cytotoxicity (phototoxicity) compared to Ce6 alone against various cancer cells. When folic acid was pretreated to block folate receptor of Y79 cells and KB cells (folate receptor-overexpressing cells), intracellular Ce6 uptake, ROS generation and thereby phototoxicity were decreased while MCF-7 cells did not significantly respond to blocking of folate receptor. These results indicated that they can be delivered by folate receptor-mediated pathway. Furthermore, in vivo pulmonary metastasis model using Y79 cells showed folate receptor specific delivery of FAPEGtaCe6 nanophotosensitizers. When folic acid was pre-administered, fluorescence intensity of lung was significantly decreased, indicating that FAPEGtaCe6 nanophotosensitizers have folate-receptor specificity in vitro and in vivo. We suggest that FAPEGtaCe6nanophotosensitizers are promising candidate for targeted PDT approach against cancer cells.
Introduction: The Authors refer to series of polymeric nanocarriers and their superiority in the area of drug delivery, but the references lead to “bulk” polymeric nanocarriers, where in fact they deal with self-organizing compounds.
Answer) Thank you for your valuable comment. According to your comment, we revised the introduction part and changed/added reference as follows:
In Introduction part
Otherwise, nanocarrier-based drug delivery systems have been spotlighted in last several decades due to their superiority in drug targeting issues [17-21]. Due to their small sizes, nanocarriers are believed to be an ideal candidate for tumor targeting [17,18]. Since surface of nanocarriers such as polymeric nanoparticles can be easily decorated with targeting moieties such as monoclonal antibody and then they can be specifically delivered to the tumor with minimization of side effects against normal cells [17,18]. Furthermore, nanocarriers based on polymeric nanoparticles or micelles can be designed to be stimulus of tumor tissues and then deliver the anticancer drug to tumor tissues specifically [18-21]. For example, Lei et al., reported that polymeric micelles having pH/temperature dual-responsive properties accelerate release rate of anticancer drug at acidic pH and higher temperature [20]. From these points of view, stimuli-responsive nanocarriers have great potentials in the tumor targeting because physiological properties of tumor microenvironment is quite different to normal counterpart, i.e. tumor tissues are characterized as acidic pH, abnormal redox-status, abundant extra- and/or intra-cellular enzymes, and elevated expression of various molecular receptors [22-25]. Therefore, abnormal status of tumor microenvironment provides tumor targeting opportunity for polymeric nanocarrier-based drug delivery [26]. Interestingly, since intracellular ROS levels in tumor tissues are significantly higher than those of normal counterpart in normal tissues or cells, these status of tumor microenvironments can be applied in nanoparticle architecture design to be sensitive to ROS level [27,28].
In reference part
- Zumaya, A.L.V.; Rimpelová, S.; Štějdířová, M.; Ulbrich, P.; Vilčáková, J.; Hassouna, F. Antibody conjugated PLGA nanocarriers and superparmagnetic nanoparticles for targeted delivery of oxaliplatin to cells from colorectal carcinoma. Int. J. Mol. Sci. 2022, 23, 1200.
- Borker, S.; Pokharkar, V. Engineering of pectin-capped gold nanoparticles for delivery of doxorubicin to hepatocarcinoma cells: an insight into mechanism of cellular uptake. Artif. Cells Nanomed. Biotechnol. 2018, 46(sup2), 826-835.
- Wu, H.C.; Kuo, W.T. Redox/pH-responsive 2-in-1 chimeric nanoparticles for the co-delivery of doxorubicin and siRNA. Polymers (Basel). 2021, 13, 4362.
- Lei, B.; Sun, M.; Chen, M.; Xu, S.; Liu, H. pH and temperature double-switch hybrid micelles for controllable drug release. Langmuir. 2021, 37, 14628-14637.
- Mamnoon, B.; Loganathan, J.; Confeld, M.I.; De Fonseka, N.; Feng, L.; Froberg, J.; Choi, Y.; Tuvin, D.M.; Sathish, V.; Mallik, S. Targeted polymeric nanoparticles for drug delivery to hypoxic, triple-negative breast tumors. ACS Appl Bio Mater. 2021, 4, 1450-1460.
Results: Title in Table 1 is not adequate to the content of the Table, and actually the entire article. The Scheme and NMR spectra presented in Figure 1 are hardly readable. The Authors state, that specific peaks are presented in the range 1-10 ppm; most of organic signals appear there, so that’s not a confirmation (not as such statement).
Answer) Thank you for your valuable comment. According to your comment, we revised the introduction part and changed/added reference as follows:
Table 1. Characterization of FAPEGtaCe6 nanophotosensitizers
|
|
Ce6 content (w/w) |
Particle size (nm)a |
|
|
Theoretical |
Experimental |
||
|
TA-sese-Ce6 FAPEGtaCe6 |
65.4 16.8 |
64.9 16.2 |
- 120.1±37.31 |
* Particle size distribution were shown in Figure 2(b).
Figure 1. Synthesis scheme and 1H NMR spectra of FAPEGtaCe6 conjugates. Synthesis scheme of TA-sese-Ce6 tetramer (a) and FAPEGtaCe6 conjugates (b). (c) 1H NMR spectra of FAPEGtaCe6 conjugates.
Authors refer to some data, which they do not show (e.g. line 116). How was the experimental and theoretical content of Ce6 established? What do you mean by dialysis procedure? Is it commonly used? Is there any reference behind it? In line 144 the Authors refer to Fig 1a, I guess they mean Fig 2a. Line 146, the Authors say aqueous solution, but in methods they describe sample preparation id DMSO/H2O mixture. TEM images are very dark, in fact this image should be repeated. Line 152 lack of unit when the size is given. Moreover, the Authors state the material is 117.1+-45.18 nm big. How did the Authors get that value? Figure 3-10 are to small and in fact unreadable. Not all description contains full explanation of the figures, e,g. in Fig 3 no explanations to a-c is given. Authors use shortcuts which in general (but not all) are explained further in the text. They should be explained at their first appearance. How does fig 5a-c explain the cell viability? What time was the MTT test performed? Line 212-13 – entire sentence is not clear. What do Authors mean by that? Which cell lines the Authors actually took to what test? It is not clear, not even after Materials and Methods explanation.
Answer) Thank you for your valuable comment.
Figure S1 was shown in the document of Supplementary materials.
The method of evaluation of experimental and theoretical contents of Ce6 was indicated in the Table 1
Table 1. Characterization of FAPEGtaCe6 nanophotosensitizers
|
|
Ce6 content (w/w) |
Particle size (nm) b |
|
|
Theoretical a |
Experimental a |
||
|
TA-sese-Ce6 FAPEGtaCe6 |
65.4 16.8 |
64.9 16.2 |
- 120.1±37.31 |
a Theoretical contents of Ce6 were calculated from molecular structure and compositions of each molecules of FAPEGtaCe6 nanophotosensitizers. For experimental contents of Ce6, nanophotosensitizers were treated with H2O2, diluted with DMSO and then measured spectrophotometrically. Ce6 contents were calculated with following equation: Ce6 content (wt.%) = (Ce6 weight/total weight of nanophotosensitizers)×100.
b Particle size distribution were shown in Figure 2(b).
Dialysis procedure means that FAPEGtaCe6 conjugates were dissolved in DMSO and then this solution was introduced into dialysis membrane following with dialysis against water to fabricate nanoparticles. Dialysis procedure is common method for fabrication of nanoparticles. We cited the reference in the Experimental section and added in the Reference section
- Jeong, Y.I.; Cheon, J.B.; Kim, S.H.; Nah, J.W.; Lee, Y.M.; Sung, Y.K.; Akaike, T.; Cho, C. Clonazepam release from core-shell type nanoparticles in vitro. J. Control. Release. 1998, 51, 169-178.
Authors say aqueous solution, but in methods they describe sample preparation id DMSO/H2O mixture: Practically, DMSO was removed by dialysis of DMSO/water mixtures. I indicated it in the experimental section.
FAPEGtaCe6 conjugates (20 mg) were dissolved in 5 ml water/DMSO mixtures (1/4, v/v) and then introduced into dialysis membrane (MWCO: 8000 g/mol) as reported previously [30,53]. This solution was dialyzed for 1 day with exchange of water every 2~3 h intervals to remove organic solvent. Resulting solution was used for analysis or PDT effect.
117.1+-45.18 nm big: We get this value from zetasizer measurement. This value means the dimeter+- size distribution. 45.18 is not a standard deviation.
As shown in Figure 3(a), nanophotosensitizers showed 117.1±45.18 nm with monomodal distribution. However, particle size distribution became broad was and multimodal pattern when H2O2 was added as shown in Figure 3(b) and (c). When 2.0 mM H2O2 was added, size distribution of nanophotosensitizers was started to become bimodal phase as shown in Figure 3(b). Especially, particle size distribution of nanophotosensitizers became bimodal distribution pattern completely by addition of 10 mM H2O2 as shown in Figure 3(c).
How does fig 5a-c explain the cell viability? What time was the MTT test performed?:
- fig 5a-c shows the Ce6 uptake ratio for 2 h. In that time, Ce6 or nanophotosensitizers were not affected to the viability of cells. When we checked cell viability at 2h after Ce6 or nanophotosensitizer treatment, viability of cells was higher than 95 % until 5 ug/ml concentration.
For MTT test, cells were exposed to Ce6 or nanophotosensitizers for 2 h and, after that, chemicals were removed by washing of cells with PBS. After that, cells were immediately irradiated and then cells were additionally cultured for 24 h. For MTT test, MTT reagent was treated cells for 3h.
Line 212-13 : Especially, ROS generation of nanophotosensitizers in Y79 cells, MCF-7 cells and KB cells was 3.2 times, 3.1 times and 3.4 times higher than those of Ce6 alone.
Discussion: Discussion is very poor in regards to the amount of results obtained. Especially when taking into consideration, that lines 285 (beginning) to 312 are general literature statements, not discussion.
Answer) Thank you for your valuable comment. According to your comment, we revised these part and added the reference.
Tumor microenvironment has deep relationship with elevated redox potential compared to normal tissues and cells [31-35]. In the tumor microenvironment, H2O2 metabolic activity is known to be decreased in tumor cells and then H2O2 in tumor microenvironment is able to be accumulated up to 100 µM, while hydrogen peroxide in normal cells is normally less than 20 nM [32-34]. Furthermore, de Sá Junior et al., reviewed that high ROS levels act as a cancer modulator and then induce genetoxic or proapoptotic effect on cancer cells [34]. They argued that these paradoxical characters of ROS guide antitumor therapeutic approaches using various chemical agents, i.e. molecules such as antioxidant chemicals can be used to prevent ROS formation and then prevent carcinogenesis. Practically, FAPEGtaCe6 nanophotosensitizers produced intracellular ROS in various cancer cells by light irradiation and then elevated ROS level induced death of cancer cells as shown in Figure 7. However, viability of cancer cells were not affected by Ce6 or nanophotosensitizer itself in the absence of light irradiation (Figure 6). Otherwise, some agents for producing ROS may use to promote a sudden increase of ROS and then kill the cancer cells through oxidative stress against tumor tissues. Low intracellular levels of ROS are required for signal transduction of normal cells while high levels of ROS in cancer cells are required to maintain their high proliferation rate and to become tumor resistance to conventional chemotherapy [35,36]. Therefore, these double-edged sword effect of ROS provides opportunity to develop therapeutic strategy, i.e. increased intracellular ROS concentration until toxic level can be applied to develop anticancer therapeutic carriers [35-37]. For example, ROS-producing agent such as piperlongumine can be used to induce apoptotic death of cancer cells via overproduction of intracellular ROS [28,38,39]. Furthermore, photosensitizers are mentioned as a typical therapeutic agent for producing ROS in cancer cells [40]. However, since conventional photosensitizers have no specificity against tumor and then potential to elevate ROS level in normal cells or tissues, its clinical application in humans is still problematic [40,41]. For example, photosensitizers act both normal cells and unhealthy cells, and then make patients sensitive to sun light for a long duration [42,43]. These drawbacks require novel delivery system for specific targeting of cancer with minimization of photosensitizers against normal cells or tissues. In our results, FAPEGtaCe6 nanophotosensitizers liberated Ce6 with ROS-sensitive manner, i.e. liberation of Ce6 from nanophotosensitizers was accelerated by the oxidative stress while Ce6 release rate was significantly lower in the absence of H2O2 than those in the presence of H2O2 as shown in Figure 4. These results must be due to the ROS-specific disintegration of nanophotosensitizers in the presence of H2O2 as shown in Figure 3. From these points of view, nanocarrier-based drug delivery system have been investigated to be sensitive against higher intracellular ROS levels [27,28,37]. For example, nanofiber mats designed to be sensitive to ROS can be apply to the site-specific release of ROS-producing agents, i.e. anticancer agent release can be accelerated from nanofiber mats with sensitive against ROS level [43].
Materials and methods: Once again, what is tetra acid? Sentence starting in line 352 is grammarly incorrect. Line 357; not to activate carboxylic acid, but carboxylic groups. Yield presented by the Authors is not actual yield. Many times the time of the reaction is not specified. What do Authors mean by “more than 24 h”? 25? 48? 200? If the reaction needs 24 h, you should state “at least 24h” etc.
Answer) Thank you for your valuable comment. In fact, we purchased tetra-acid from Frontier Scientific (https://www.frontiersci.com/product/tetraacid/). As you indicated, tetra acid is not exact chemical name and therefore its chemical name was indicated in the Introduction/experimental section. Furthermore, when we used the term “24 h”, we practically reacted the reactants for 24 h. Then we corrected them as you indicated. “Yield” part was also changed to Yield of FA-PEG-CoOOH to yield of lyophilized solid.
4.1. Materials
3-[3-(2-carboxyethoxy)-2,2-bis(2-carboxyethoxymethyl)propoxy]propanoic acid (tetra acid, TA)
The dialysis membranes having molecular weight cut-off (MWCO) of 1000, 2000 and 8000 g/mol were purchased from Spectra/ProTM Membranes.
This solution was stirred for 3 h to activate carboxyl group of FA.
The yield of lyophilized solid was calculated as followed: Yield = [weight of lyophilized solid/(weight of FA + weight of PEG)]*100. The yield of lyophilized solid was higher than 97 % (w/w).
Line 408 – this should be a new subsection. Line 411: diluted more than 10 times? Section 4.8. no explanation of cell lines is given. Sections 4.9, 4.10 and 4.11 which cell lines?
Conclusions: needs to be rewritten.
Answer) Thank you for your valuable comment. According to your comment, we revised the manuscript
4.5. Ce6 contents in the FAPEGtaCe6 nanophotosensitizers
Contents of Ce6 in the conjugates was estimated as follows: FAPEGtaCe6 nanophotosensitizers fabricated as described above were adjusted to 1 mg/ml and then 5 ml this solution was mixed with 45 ml phosphate buffereed saline (PBS, 0.01 M, pH 7.4) in the presence of 20 mM H2O2. This solution was stirred for 48 h and then diluted with DMSO more than ten times. Ce6 concentration was measured with an Infinite M200 pro microplate reader (Tecan) (excitation wavelength: 407, emission wavelength: 664 nm). Ce6 itself dissolved in DMSO was diluted 20 times with PBS (20mM H2O2) was similarly used instead of nanophotosensitizer solution for standard test.
Ce6 content (wt.%) = (Ce6 weight/total weight of nanophotosensitizers)×100.
Ce6 contents were about 16.2 % (w/w).
4.9 cell culture
Y79 retinoblastoma cells and ARPE-19 human retinal pigment epithelial cells were purchased from American Type Culture Collection (ATCC, Manassas, VA USA). KB human epithelial and MCF-7 human breast cancer cells were obtained from Korean Cell Line Bank (Seoul, Korea).
4.10. PDT treatment
Y79, MCF-7 and KB cells (2×104 cells/well) were seeded in 96 well plates, respectively, and then exposed to Ce6 or nanophotosensitizers.
For dark toxicity, ARPE-19, Y79, MCF-7 and KB cells (2×104 cells/well) were seeded in 96 well plates, respectively, and then similarly treated as described above without light irradiation.
To summarize, the Authors need to carefully read and improve their article before it can be accepted.
Answer) Thank you for your valuable comment. According to your comment, we corrected the grammatical error by aid of commercial English editing services.

Reviewer 2 Report
Dear Authors,
This manuscript presents exciting and novel folic-acid targeted and ROS-sensitive nanoparticles for photodynamic therapy. Although the subject is fascinating, the language and text structure hamper the reader's understanding. I would strongly recommend revising the text carefully or having a native speaker do it.
- The in vitro study was carried out with a single light dose of 2 J/cm2 and photosensitizer concentration, reaching cell-kill rates up to 90%. Is there any reason not to evaluate multiple light doses and photosensitizer concentrations and determine LD50 and IC50?
- Add to or comment in the text whether the Chlorin e6 encapsulation (nanoparticle) affects the Chlorin e6 absorption spectrum. If it does, then adding the spectrum is recommended.
- Are the H2O2 concentrations tested in vitro equivalent to the tumor intracellular ROS?
- Add the laser intensity and irradiation time.
- How was the drug-light interval determined? In Fig 8, a low Ce6 fluorescence signal is observed in the cells. Does it change over time? If the nanoparticles and Ce6 alone were incubated for 4h, would the uptake still be the same? I recommend repeating the fluorescence images at different time points to optimize both photosensitizers' uptake and then performing the PDT study with this optimized incubation time.
- Add the Maestro imaging protocol (excitation wavelength, filters..).
- It is not clear why the nanoparticle-dependence disintegration to H2O2 would improve the PDT response as stated in the discussion: "Furthermore, in our results, nanophotosensitizers may provide synergistic effects on the Ce6-based PDT, i.e. nanophotosensitizers can be specifically delivered to the site of high oxidative stress and, thereafter, disintegration of nanophotosensitizers by ROS may accelerate Ce6-mediated ROS production and PDT efficacy."
- Is there any significant difference in the levels of folic acid expressed by the cells studied? If yes, does the photosensitizer uptake correspond to this? What about the ROS?
Author Response
Response to reviewer 2’s comment
This manuscript presents exciting and novel folic-acid targeted and ROS-sensitive nanoparticles for photodynamic therapy. Although the subject is fascinating, the language and text structure hamper the reader's understanding. I would strongly recommend revising the text carefully or having a native speaker do it.
Answer) Thank you for your valuable comment. According to your comment, we fully revised the manuscript and grammatical errors were corrected by aid of native speaker of commercial English Editing Services. Thank again.
- The in vitrostudy was carried out with a single light dose of 2 J/cm2 and photosensitizer concentration, reaching cell-kill rates up to 90%. Is there any reason not to evaluate multiple light doses and photosensitizer concentrations and determine LD50 and IC50?
Answer) Thank you for your valuable comment. Practically, our group have been already tried to determine light dose as a preliminary study using Ce6. In that study, we decided that 2.0 J/cm2 is appropriate for treatment of various kind of cells. Even though the higher the light dose induces the higher the cell cytotoxicity, higher light dose requires increased exposure time of cells in the normal air and room temperature (because light irradiation devices are in the normal air and room temperature. Due to its size, we have difficulties to put inside of CO2 incubator.). Then, we decided 2.0 J/cm2 is appropriate for treatment of various kind of cells. Furthermore, at higher dose than 2.0 J/cm2, viability of normal cells was also affected by light irradiation. For example, at 5.0 J/cm2, and 5 µg/ml Ce6 concentration, the viability of CCD986Sk cells was lower than 70 % (this data is not shown in this paper) even though viability of cancer cells was also decreased less than 70 %. Other problem is temperature, i.e. when light dose was higher than 3.0 J/cm2, temperature in the well plate was increased and then affected to the cell viability. Practically, at higher than 10 J/cm2, temperature was increased (higher than 38 oC) and viability of cells must be affected by increased temperature.
Anyway, we added preliminary data about light dose on the cell viability in the supporting materials and then discussed more about this in the Results section.
In Results section
Prior to perform PDT study, the effect of light dose against of Y79 cells was evaluated. As shown in Figure S2, the viability of Y79 cells were gradually decreased according to the increase of light dose. The viability of Y79 cells were not practically affected until 1.5 J/cm2, i.e. cell viability was higher than 90 % at 1.5 J/cm2 and then cell viability was gradually decreased until 20 J/cm2. However, the temperature was also raised when light dose was higher than 5 J/cm2 (data not shown) and then this may affect to the viability of cells. To minimize, 2 J/cm2 was used for PDT study.
In supplementary information
Figure S2. The effect of light dose on the viability of Y79 cells. Ce6 concentration was 2 µg/ml. Y79 cells (2×104 cells/well) seeded in 96 well plate was cultured in 5 % CO2 incubator at 37oC overnight. Following this, Ce6 2 (µg/ml) were treated and then irradiated with various light dose using expanded homogenous beam (SH Systems, Gwangju, Korea). 24 h later, viability of cells was evaluated y MTT assay. Viability of cell at light dose of 0 J/cm2) was set 100 % as a control and then compared.
Table 2. The effect of Ce6 alone or FAPEGtaCe6 nanophotosensitizers on the IC50 value
|
|
IC50 (ug/ml) |
|
|
Ce6 |
FAPEGtaCe6 NPT |
|
|
Y79 MCF-7 KB |
> 6.4 4.93 > 5.1 |
0.16 0.29 0.19 |
* IC50 values were derived from Figure 7B.
- Add to or comment in the text whether the Chlorin e6 encapsulation (nanoparticle) affects the Chlorin e6 absorption spectrum. If it does, then adding the spectrum is recommended.
Answer) Thank you for your valuable comment. According to your comment, we added the UV-VIS spectrum in the Figure 2 and the explanation was added to Results section.
UV absorption spectra of Ce6 alone and FAPEGtaCe6 nanophotosensitizers were measured as shown in Figure 2(a). As shown in Figure 2(a), FAPEGtaCe6 nanophotosensitizers represented similar UV absorption spectra between 600 nm ~ 700 nm in DMSO. Furthermore, specific peak at 664 nm was also observed both of Ce6 alone and FAPEGtaCe6 nanophotosensitizers, indicating that intrinsic absorption properties of Ce6.
Figure 2. (a) UV absorption of Ce6 or FAPEGtaCe6 nanophotosensitizers in DMSO (0.1 mg/ml Ce6 equivalent). (b) TEM image and (c) particle size distribution of FAPEGtaCe6 nanophotosensitizers. (d) Stability of FAPEGtaCe6 nanophotosensitizers. For stability test, aqueous nanophotosensitizer solution (1 mg FAPEGtaCe6 nanophotosensitizers/ml deionized water) was left in the room temperature (20oC). (e) Singlet oxygen generation from Ce6 alone or FAPEGtaCe6 nanophotosensitizers under light irradiation at 664 nm (n = 4).
Materials and methods section
UV absorption spectra of Ce6 or FAPEGtaCe6 nanophotosensitizers were measured with Genesys 10s UV-VIS spectrophotometer (Thermo Fisher Scientific, Waltham, Massachusetts, USA).
- Are the H2O2 concentrations tested in vitro equivalent to the tumor intracellular ROS?
Answer) Thank you for your valuable comment. Practically, H2O2 concentration in tumor microenvironment was reported to 50–100 μM. However, we tested the ROS-sensitivity of nanophotosensitizers in the higher H2O2 concentration than that of tumor microenvironment to know whether or not they responded to ROS. We discussed more about these events in the discussion section as follows:
In this study, we used higher H2O2 concentrations than that of tumor microenvironment to get clear on the ROS-sensitivity of nanophotosensitizers. As shown in Figure 3 and 4, H2O2 addition in the nanophotosensitizer solution induced modulation of particle size distribution. Pandya et al., also reported that paclitaxel-incorporated nanoparticles were degraded by incubation with 5 mM H2O2 and then particle size was modified [47]. Our group also previously reported that poly(DL-lactide-coglycolide)/poly(ethylene glycol) nanoparticles having diselenide linkages showed H2O2-dependent degradation and then antibiotic release rate was dependent on the H2O2-concentration, i.e. nanoparticles degradation was accelerated at 10 mM H2O2 rather than 0 or 1 mM H2O2 and then ciprofloxacin release rate also accelerated [48]. This study also showed that Ce6 release rate was significantly increased when 10 mM H2O2 was added to the release media as shown in Figure 4. These results indicated that FAPEGtaCe6 nanophotosensitizers have ROS-sensitivity and can be responded to the oxidative stress of tumor microenvironment.
- Pandya, A.D.; Jäger, E.; Bagheri Fam, S.; Höcherl, A.; Jäger, A.; Sincari, V.; Nyström, B.; Štěpánek, P.; Skotland, T.; Sandvig, K.; Hrubý, M.; Mælandsmo, G.M. Paclitaxel-loaded biodegradable ROS-sensitive nanoparticles for cancer therapy. Int J Nanomedicine. 2019, 14, 6269-6285.
- Song, J.; Kook, M.S.; Kim, B.H.; Jeong, Y.I.; Oh, K.J. Ciprofloxacin-releasing ROS-sensitive nanoparticles composed of poly(ethylene glycol)/poly(D,L-lactide-co-glycolide) for antibacterial treatment. Materials (Basel). 2021, 14, 4125.
- Add the laser intensity and irradiation time.
Answer) Thank you for your valuable comment. According to your comment, we indicated the laser intensity and irradiation time in the Experimental section.
The light intensity of expanded homogenous beam and treatment time was 3.515 mW/cm2 and 569 s (2000 mJ/cm2).
- How was the drug-light interval determined? In Fig 8, a low Ce6 fluorescence signal is observed in the cells. Does it change over time? If the nanoparticles and Ce6 alone were incubated for 4h, would the uptake still be the same? I recommend repeating the fluorescence images at different time points to optimize both photosensitizers' uptake and then performing the PDT study with this optimized incubation time.
Answer) Thank you for your valuable comment. As you indicated, I agree to your comment, i.e. increase in incubation time must be increased in intracellular fluorescence intensity by treatment of Ce6 alone because Ce6 is a hydrophobic agent and then it efficiently enters into the intracellular compartment. In this study, we wanted to show the efficacy of nanophotosensitizers for intracellular delivery against cancer cells with folate receptor-mediated pathway. If the nanoparticles and Ce6 alone were incubated for 4h, uptake must become similar. Anyway, as you indicated, we treated Ce6 alone and nanoparticles for 4 h and then compared in the supplementary files.
Furthermore, intracellular uptake of Ce6 or nanophotosensitizers were gradually increased according to the treatment time, i.e. when Ce6 or nanophotosensitizers were exposed to cells for 4 h, intracellular uptake ratio was increased both of Ce6 alone and nanophotosensitizers as shown in Figure S3. Especially, intracellular uptake ratio of Ce6 alone for 4 h was increased more than 2 times compared to that of 1.5 h while nanophotosenstizer treatment increased 30 % in intracellular uptake ratio compared to 1.5 h. These results indicated that Ce6 uptake rate of nanophotosensitizers was higher and faster than that of Ce6 alone even though Ce6 uptake ratio of Ce6 alone was also gradually increased according to the time course.
Figure S3. The effect of treatment time of Ce6 and nanophotosensitizers on the intracellular Ce6 uptake of KB cells. (a) Fluorescence observations (b) intracellular Ce6 uptake against various cancer cells (b). Ce6 concentration was 2 µg/mL. Bar = 20 µm.
- Add the Maestro imaging protocol (excitation wavelength, filters..).
Answer) Thank you for your valuable comment. As you indicated, we added the protocol of Maestro imaging tools (excitation wavelength and filters) in the experimental section.
In experimental section (4.12. In vivo animal tumor imaging)
For fluorescence observation of each organs, filter of MaestroTM 2 small animal imaging instrument was Orange filter set (Excitation filter: 586-631 nm, Emission filter: 645 nm longpass).
- It is not clear why the nanoparticle-dependence disintegration to H2O2 would improve the PDT response as stated in the discussion: "Furthermore, in our results, nanophotosensitizers may provide synergistic effects on the Ce6-based PDT, i.e. nanophotosensitizers can be specifically delivered to the site of high oxidative stress and, thereafter, disintegration of nanophotosensitizers by ROS may accelerate Ce6-mediated ROS production and PDT efficacy."
Answer) Thank you for your valuable comment. At this moment, this paragraph is to emphasize the ROS-sensitive disintegration of nanophotosensitizers must be accelerated by PDT treatment and then increase in Ce6 liberation would also accelerate PDT efficacy because PDT effect against cancer cells is realized when Ce6 liberated from nanophotosensitizers. Therefore, if H2O2 accelerate Ce6 release from nanophotosensitizers, it would accelerate Ce6 uptake and ROS production in cancer cells. Anyway, to avoid confuse complexity of this phrases, we revised the manuscript and discussed again as follows:
In our results, nanophotosensitizers can be disintegrated in the presence of H2O2 and then Ce6 release rate from nanophotosensitizers can be also accelerated as shown in Figure 4. These results indicated that nanophotosensitizers can be specifically delivered to the disease site having high oxidative stress and, thereafter, liberated Ce6 can be preferentially delivered to disease cell. Since oxidative stress is normally elevated in cancer cells, these peculiarities of cancer cells can be applied in targeting of anticancer agents [38,46].
- Hayes, J.D.; Dinkova-Kostova, A.T.; Tew, K.D. Oxidative stress in cancer. Cancer Cell. 2020, 38, 167-197.
- Is there any significant difference in the levels of folic acid expressed by the cells studied? If yes, does the photosensitizer uptake correspond to this? What about the ROS?
Answer) Thank you for your valuable comment. Because expression level of folate receptor on the cancer cell membrane is different along with cancer cell type. For example, MDA-MB 231 cells (this is also breast cancer cells as well as MCF7 cells) is known to express folate receptor with high degree while MCF7 cells expresses relatively folate receptors with lower degree. As you indicated, increased intracellular ROS level and/or tumor-microenvironment ROS level must be also inducer for targeting of cancer cells. Lee et al. reported that H2O2 addition into the culture medium of cancer cells increases intracellular delivery nanoparticles and improves anticancer activity. Therefore, elevated ROS level is also a inducer for drug targeting [27]. Anyway, we discussed more about this in the Results and Discussion section.
Additionally, H2O2 was added to the culture medium to investigate the effect of extracellular ROS on the Ce6 uptake ratio. When H2O2 was added to the FA-pretreatment group, fluorescence intensity of Y79 cells and KB cells was increased while fluorescence intensity of Ce6 treatment was not significantly changed as shown in Figure 8(a) and (b). These results indicated that FAPEGtaCe6 nanophotosensitizers can be delivered to the cancer cells via ROS-sensitive mechanisms.
Figure 8. Fluorescence observations of KB cells (a) and intracellular Ce6 uptake against various cancer cells (b). FA (5.0 mM) was pretreated to the KB cells 30 min before treatment of Ce6 or nanophotosensitizers. For H2O2 (2 mM) treatment, Ce6 or nanophotosensitizers were treated to cells and then H2O2 was added to the medium. Ce6 concentration was 2 µg/mL. Bar = 20 µm.

Reviewer 3 Report
Comments:
Interesting and well-conducted work but it needs some major review in some parts:
- Please improve the resolution of the figures in general.
- Please provide the absorption spectra of nanophotosensitizer.
- Please provide the intensity, volume, and size graphs of DLS in the Supporting information.
- The overall ROS detection measured by DCFH-DA is insufficient. Authors should detect various ROS including at least hydroxyl radicals and singlet oxygen. Please provide the different ROS detection experiments (e.g. using fluorescence probes (like APF, DHE, SOSG, HPF), EPR or singlet oxygen phosphorescence) and analyze the mechanisms of ROS generation by investigated nanophotosensitizers (type I vs type II) and compared these data to Ce6 (unmodified).This experiment should be performed both in solution and in cells.
- Throughout the paper the authors only elaborate on the cancer cell lines. Their main take home message is, that tested photosensitizer selectively kills cancer cells (folate receptor targeting). Therefore it is necessary to repeat all the experiments with a non-cancerous cell line (e.g. HaCat) that then serves as a negative control. This is one of the major points of criticism and non-negotiable to me.
- The expression of folate receptor in selected cell lines has to be confirmed and validated eg. by flow cytometry analysis.
- The uptake presented in Fig.5 should be quantified and the data have to be presented as PS concentration per cell. The fluorescence microscopy pictures are not focused. Moreover on bright field images were captured with blue filter (blue background). The scale bars are invisible. Authors have to provide images with much more better resolution and quality.
- Fig. 8A - as above, the fluorescence microscopy pictures are not focused and have to be corrected.
- The phototoxicity test in vitro has to be included.
- Did Authors check the long term antitumor effect mediated by investigated photosensitizer?
- The statistical analysis is missing.
Author Response
Response to Reviewer 3’s comment
- Please improve the resolution of the figures in general.
Answer) Thank you for your valuable comment. According to your comment, we improved the resolution of all the figures as far as we can.
- Please provide the absorption spectra of nanophotosensitizer.
Answer) Thank you for your valuable comment. According to your comment, we added the UV-VIS spectrum in the Figure 2 and the explanation was added to Results section.
UV absorption spectra of Ce6 alone and FAPEGtaCe6 nanophotosensitizers were measured as shown in Figure 2(a). As shown in Figure 2(a), FAPEGtaCe6 nanophotosensitizers represented similar UV absorption spectra between 600 nm ~ 700 nm in DMSO. Furthermore, specific peak at 664 nm was also observed both of Ce6 alone and FAPEGtaCe6 nanophotosensitizers, indicating that intrinsic absorption properties of Ce6.
Figure 2. (a) UV absorption of Ce6 or FAPEGtaCe6 nanophotosensitizers in DMSO (0.1 mg/ml Ce6 equivalent). (b) TEM image and (c) particle size distribution of FAPEGtaCe6 nanophotosensitizers. (d) Stability of FAPEGtaCe6 nanophotosensitizers. For stability test, aqueous nanophotosensitizer solution (1 mg FAPEGtaCe6 nanophotosensitizers/ml deionized water) was left in the room temperature (20oC). (e) Singlet oxygen generation from Ce6 alone or FAPEGtaCe6 nanophotosensitizers under light irradiation at 664 nm (n = 4).
Materials and methods section
UV absorption spectra of Ce6 or FAPEGtaCe6 nanophotosensitizers were measured with Genesys 10s UV-VIS spectrophotometer (Thermo Fisher Scientific, Waltham, Massachusetts, USA).
- Please provide the intensity, volume, and size graphs of DLS in the Supporting information.
Answer) Thank you for your valuable comment. According to your comment, we presented the volume and size results was added in the supporting information.
Table S2. The effect of H2O2 addition on the particle size distribution of FAPEGtaCe6 nanophotosensitizers
|
H2O2 (mM) |
Particle size distribution (nm) |
|||
|
Intensity |
Polydispersity index (PDI) |
Volume |
Polydispersity index (PDI) |
|
|
0 |
117.1±45.18 |
0.244 |
127.4±47.79 |
0.228 |
|
2 |
166.8±68.87 (83.6 %) 40.98±10.06 (16.4 %) |
0.279 |
169.6±72.56 (84.9 %) 37.76±10.67 (15.1 %) |
0.280 |
|
10 |
531.5±183.8 (76.9 %) 97.91±22.07 (23.1 %) |
0.461 |
425.8±162.7 (87.2) 75.67±16.07 (12.8) |
0.315 |
* Particle sizes and PDI indexes were Figure 3 (Intensity %) and Figure S2 (Volume %).
Figure S2. The effect of H2O2 concentration on the changes of particle size distribution (volume %). To study ROS sensitivity, nanophotosensitizers in PBS (1 mg/ml) was incubated in the presence of H2O2 at 37oC for 3 h.
- The overall ROS detection measured by DCFH-DA is insufficient. Authors should detect various ROS including at least hydroxyl radicals and singlet oxygen. Please provide the different ROS detection experiments (e.g. using fluorescence probes (like APF, DHE, SOSG, HPF), EPR or singlet oxygen phosphorescence) and analyze the mechanisms of ROS generation by investigated nanophotosensitizers (type I vs type II) and compared these data to Ce6 (unmodified). This experiment should be performed both in solution and in cells.
Answer) Thank you for your valuable comment. According to your comment, we checked singlet oxygen generation by ce6 alone and nanophotosensitizers in aqueous solution using SOSG reagent. At this moment, we are now going to do various ROS formation in the cells using various cancer cells and normal cells. We hope these results in the separated papers. Anyway, we presented SO measurement results in solutions in Figure 2(d) and discussed more in the results section. Thanks again for your good comment.
Singlet oxygen (SO) generation efficiency of nanophotosensitizers was evaluated in aqueous condition using SOSG reagent as shown in Figure 2(e). Fluorescence intensity of SOSG was gradually increased according to the increase of irradiation time both of Ce6 alone and FAPEGtaCe6 nanophotosensitizers. Especially, fluorescence intensity of nanophotosensitizers was higher than Ce6 alone. Furthermore, fluorescence intensity of nanophotosensitizers was not significantly increased in the absence of light irradiation. These results indicated that FAPEGtaCe6 nanophotosensitizers efficiently produced successfully generated SO in the aqueous solution.
Figure 2. (a) UV absorption of Ce6 or FAPEGtaCe6 nanophotosensitizers in DMSO (0.1 mg/ml Ce6 equivalent). (b) TEM image and (c) particle size distribution of FAPEGtaCe6 nanophotosensitizers. (d) Stability of FAPEGtaCe6 nanophotosensitizers. For stability test, aqueous nanophotosensitizer solution (1 mg FAPEGtaCe6 nanophotosensitizers/ml deionized water) was left in the room temperature (20oC). (e) Singlet oxygen generation from Ce6 alone or FAPEGtaCe6 nanophotosensitizers under light irradiation at 664 nm (n = 4).
In Materials and methods section
SO generation from Ce6 or nanophotosensitizers in aqueous solution was measured using 1 mL of Ce6 alone or nanophotosensitizer solution (5 μg/mL Ce6 equivalent in distilled water, 1% DMSO) [55]. To this solution, SOSG reagent was added (final concentration: 5 µM). Each solution was irradiated with expanded homogenous beam (664 nm, SH Systems, Gwangju, Korea) at different time points (0.5, 1, 2, 5, 10 min). After that, fluorescence intensity of this solution was measured with fluorescence spectrophotometer (RF-5301PC, Shimadzu Co., Kyoto, Japan) at 488 nm of excitation wavelength and 525 nm of emission wavelength. This procedure was carried out under dark conditions.
- Throughout the paper the authors only elaborate on the cancer cell lines. Their main take home message is, that tested photosensitizer selectively kills cancer cells (folate receptor targeting). Therefore, it is necessary to repeat all the experiments with a non-cancerous cell line (e.g. HaCat) that then serves as a negative control. This is one of the major points of criticism and non-negotiable to me.
Answer) Thank you for your valuable comment. In fact, photosensitizer itself does not have specificity against cancer cells and/or non-cancerous normal cells. We already did dark-toxicity using non-cancerous ARPE-19 cells, which is a non-cancerous cells (retinal cells) as shown in Figure 6(a). However, at in vitro study, nanophotosensitizers will also represent against normal cells such as ARPE-19 and HaCaT cells because FAPEGtaCe6 nanophotosensitizers can be also delivered to normal cells with passive delivery mechanism (such as endocytosis), i.e. nanophotosensitizers can be entered into the cells by plain endocytosis mechanism and they will affect to the cells by light irradiation. Anyway, we presented Ce6 uptake ratio, ROS formation, Dark toxicity and phototoxicity in Figure S5 and more discussed in the Results section.
Figure 6. Dark toxicity of free Ce6 and nanophotosensitizers. Normal cell lines such as ARPE-19 cells (a) and HaKaT cells (b). Cancer cell lines such as Y79 cells (c), MCF-7 cells (d) and KB cells (e). NPT = FAPEGtaCe6 nanophotosensitizers.
Figure S5. The effect of Ce6 and nanophotosensitizers against non-cancerous normal cell lines. (a) Intracellular Ce6 uptake ratio; (b) intracellular ROS generation; (c) Phototoxicity. All experiments for normal cells were carried out as similar to cancer cells.
In our results using non-cancerous normal cell lines such as ARPE-19 and HaKaT cells (Figure S5), FAPEGtaCe6 nanophotosensitizers showed higher intracellular ce6 uptake (Figure S5(a)), ROS generation (Figure S5(b)) and phototoxicity (Figure S5(c)) as similar to cancer cells even though their gap was smaller than those of cancer cells. These results mean that nanophotosensitizers are also able to be delivered to normal cells with non-specific manner. However, in vivo study using pulmonary metastasis showed that nanophotosensitizers were specifically delivered to Y79 cell-metastasized lung as shown in Figure 10. These results indicated that FAPEGtaCe6 nanophotosensitizers can be delivered to tumor cells in the biological system of human body. Huang et al., also reported that folic acid-conjugated graphene oxide significantly increases the accumulation of Ce6 in tumor cells and lead to a remarkable photodynamic efficacy on MGC803 cells upon irradiation [52].
- The expression of folate receptor in selected cell lines has to be confirmed and validated eg. by flow cytometry analysis.
Answer) Thank you for your valuable comment. Practically, folate receptor expression of Y79 cells and KB cells already reported by other scientists and expression of folate receptor of MCF-7 cells is less than these cell lines. Anyway, I discussed more in the discussion section and cited more references.
For example, folate receptors are overexpressed in malignant cells such as Y79 human retinoblastoma cells compared to normal cells such as ARPE-19 cells [50]. This event also can be applied to develop nanomedicine to be sensitive to elevated ROS level. Alsaab et al., reported that folic acid-decorated nanomicelles represent improved cytotoxicity against retinoblastoma cells while folate receptor negative normal cells such as ARPE-19 cells was not affected by nanomicelles [51].
- Jwala, J.; Vadlapatla, R.K.; Vadlapudi, A.D.; Boddu, S.H.; Pal, D.; Mitra, A.K. Differential expression of folate receptor-alpha, sodium-dependent multivitamin transporter, and amino acid transporter (B (0, +)) in human retinoblastoma (Y-79) and retinal pigment epithelial (ARPE-19) cell lines. J. Ocul. Pharmacol. Ther. 2012, 28, 237-44.
- Alsaab, H.; Alzhrani, R.M.; Kesharwani, P.; Sau, S.; Boddu, S.H.; Iyer, A.K. Folate decorated nanomicelles loaded with a potent curcumin analogue for targeting retinoblastoma. Pharmaceutics. 2017, 9, 15.
- Feng, D.; Song, Y.; Shi, W.; Li, X.; Ma, H. Distinguishing folate-receptor-positive cells from folate-receptor-negative cells using a fluorescence off-on nanoprobe. Anal. Chem. 2013, 85, 6530-6535.
- The uptake presented in Fig.5 should be quantified and the data have to be presented as PS concentration per cell. The fluorescence microscopy pictures are not focused. Moreover on bright field images were captured with blue filter (blue background). The scale bars are invisible. Authors have to provide images with much more better resolution and quality.
Answer) Thank you for your valuable comment. According to your comment, we performed experiment of Ce6 uptake again and expressed as PS concentration per mg protein in Figure 5(a)-(c). (Due to COVID-19, the lab of analytical devices were closed. Then we have some difficulties to use fluorescence microscopy). As you indicated, images of Figure 5(d) is adjusted to show better. Scale bar was added in the Figures. Thanks again.
Figure 5. Ce6 uptake of Y79 cells (a), MCF-7 cells (b) and KB cells (c). Fluorescence observation of KB cells with treatment of Ce6 or nanophotosensitizers (d). Bar = 20 µm.
- Fig. 8A - as above, the fluorescence microscopy pictures are not focused and have to be corrected.
Answer) Thank you for your valuable comment. As you indicated, images of Figure 5(d) is adjusted to show better. Scale bar was added in the Figures. Thanks again.
Figure 8. Fluorescence observations of KB cells (a) and intracellular Ce6 uptake against various cancer cells (b). FA (5.0 mM) was pretreated to the KB cells 30 min before treatment of Ce6 or nanophotosensitizers. For H2O2 (2 mM) treatment, Ce6 or nanophotosensitizers were treated to cells and then H2O2 was added to the medium. Ce6 concentration was 2 µg/mL. Bar = 20 µm. *,**: p < 0.01.
- The phototoxicity test in vitro has to be included.
Answer) Thank you for your valuable comment. Figure 9(b) is a phototoxicity and we revised the manuscript according to your comment.
Figure 9. The effect of pretreatment of FA on the intracellular ROS generation (a) and phototoxicity (b). Cells were treated with Ce6 or nanophotosensitizers as similar to Figure 8. FA (5 mM) was pretreated to the cells 30 min before free Ce6 or nanophotosensitizer treatment. Ce6 concentration was 2 µg/mL. Cells were exposed to free Ce6 or nanophotosensitizers for 2 h and then irradiated at 664 nm (2 J/cm2). *,**: p < 0.01.
- Did Authors check the long term antitumor effect mediated by investigated photosensitizer?
Answer) Thank you for your valuable comment. At this moment, we could not perform long term antitumor effect because animal breeding center has some problem and closed last month due to the COVID-19. As far as we can, we will perform long term animal experiment in the next time and will report other article. Thanks for your comment. I appreciated.
- The statistical analysis is missing.
Answer) Thank you for your valuable comment. According to your comment, we added the statistical analysis in the Figures.
4.12. Statistical Analysis
Student's t test using SigmaPlot® program (SigmaPlot® v.11.0, Systat Software, Inc., San Jose, CA, USA) was used to estimate the statistical significance of the results and evaluate p < 0.05 as the minimal level of significance.
Figure 8. Fluorescence observations of KB cells (a) and intracellular Ce6 uptake against various cancer cells (b). FA (5.0 mM) was pretreated to the KB cells 30 min before treatment of Ce6 or nanophotosensitizers. For H2O2 (2 mM) treatment, Ce6 or nanophotosensitizers were treated to cells and then H2O2 was added to the medium. Ce6 concentration was 2 µg/mL. Bar = 20 µm. *,**: p < 0.01.
Figure 9. The effect of pretreatment of FA on the intracellular ROS generation (a) and phototoxicity (b). Cells were treated with Ce6 or nanophotosensitizers as similar to Figure 8. FA (5 mM) was pretreated to the cells 30 min before free Ce6 or nanophotosensitizer treatment. Ce6 concentration was 2 µg/mL. Cells were exposed to free Ce6 or nanophotosensitizers for 2 h and then irradiated at 664 nm (2 J/cm2). *,**: p < 0.01.

Round 2
Reviewer 3 Report
In general, please increase the quality of the figures.
Author Response
Answer to Reviewer 3’s comment.
In general, please increase the quality of the figures.
Answer) Thanks for your comment. According to your comment, we improved the quality of Figures. Thanks against.
